# Magnetic Preference Optimization: Achieving Last-iterate Convergence for Language Model Alignment

**Mingzhi Wang**[1,2], **Chengdong Ma**[1], **Qizhi Chen**[1], **Linjian Meng**[3], **Yang Han**[4]
**Jiancong Xiao**[5], **Zhaowei Zhang**[1], **Jing Huo**[3], **Weijie J. Su**[5], **Yaodong Yang**[1]*

[1]Institute for Artificial Intelligence, Peking University
[2]Beijing Academy of Artificial Intelligence
[3]National Key Laboratory for Novel Software Technology, Nanjing University
[4]China Telecom, [5]University of Pennsylvania

## ABSTRACT

Self-play methods have demonstrated remarkable success in enhancing model capabilities across various domains. In the context of Reinforcement Learning from Human Feedback (RLHF), self-play not only boosts Large Language Model (LLM) performance but also overcomes the limitations of traditional Bradley-Terry (BT) model assumptions by finding the Nash equilibrium (NE) of a preference-based, two-player constant-sum game. However, existing methods either guarantee only average-iterate convergence, incurring high storage and inference costs, or converge to the NE of a regularized game, failing to accurately reflect true human preferences. In this paper, we introduce Magnetic Preference Optimization (MPO), a novel approach capable of achieving last-iterate convergence to the NE of the original game, effectively overcoming the limitations of existing methods. Building upon Magnetic Mirror Descent (MMD), MPO attains a linear convergence rate, making it particularly suitable for fine-tuning LLMs. To ensure our algorithm is both theoretically sound and practically viable, we present a simple yet effective implementation that adapts the theoretical insights to the RLHF setting. Empirical results demonstrate that MPO can significantly enhance the performance of LLMs, highlighting the potential of self-play methods in alignment.

## 1 INTRODUCTION

Self-play has emerged as an effective method for improving model performance, particularly in domains that require strategic decision-making and complex problem-solving (Silver et al., 2017; Vinyals et al., 2019; Perolat et al., 2021). By allowing models to iteratively refine their strategies through self-competition, self-play enables them to discover optimal policies. In the realm of Reinforcement Learning from Human Feedback (RLHF) (Ouyang et al., 2022; Peng et al., 2023; Achiam et al., 2023; Jin et al., 2025), self-play not only has proven effective in enabling Large Language Models (LLMs) to better align with human preferences (Chen et al., 2024; Wu et al., 2024; Zhang et al., 2024), but also offers unique advantages by addressing the limitations of traditional preference modeling methods (Munos et al., 2023; Swamy et al., 2024).

Conventional RLHF methods typically rely on the Bradley-Terry (BT) (Bradley & Terry, 1952) assumption for preference modeling, which presumes transitivity in human preferences—if response *A* is preferred over *B*, and *B* over *C*, then *A* should also be preferred over *C*. While this may hold for individuals in specific contexts, generalizing transitive preferences across broader populations often fails due to the presence of non-transitive preferences (Swamy et al., 2024). This limitation undermines the ability of existing RLHF methods to capture the complexity of human preferences. Self-play, however, offers a solution by finding the Nash equilibrium (NE) of a two-player constant-sum game based on human preferences (Munos et al., 2023; Swamy et al., 2024).

---

*Correspondence to: Yaodong Yang <yaodong.yang@pku.edu.cn>

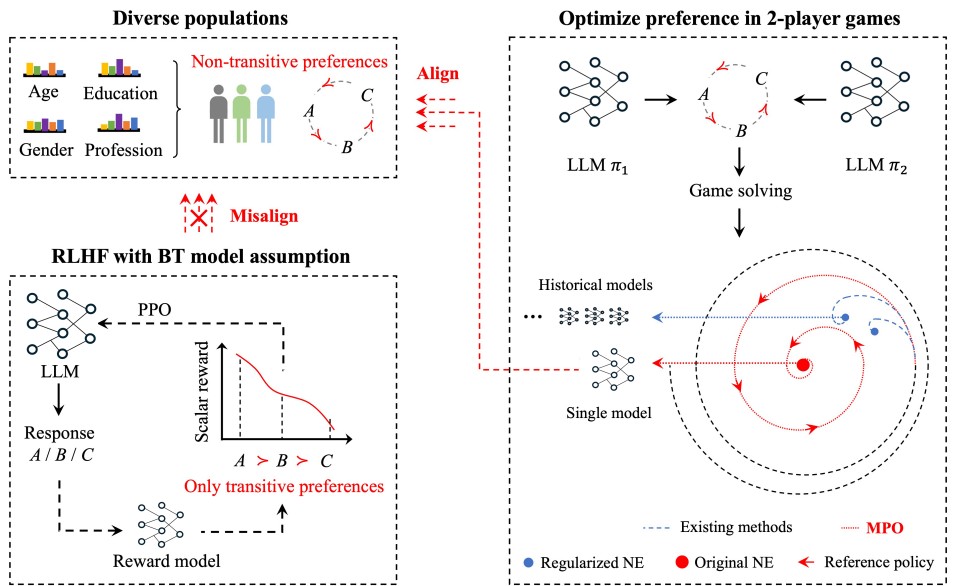

Figure 2: An illustration of MPO and its background. Non-transitive preferences are prevalent across diverse populations, necessitating a more generalized preference model that frames the alignment problem as a two-player constant-sum game. Existing methods either converge to the NE of a regularized game or require maintaining multiple models. In contrast, MPO achieves last-iterate convergence to the original NE, aligning with diverse human preferences using only a single model.

Despite its promise, self-play in the context of LLM alignment presents unique challenges. Most existing methods, such as Self-Play Preference Optimization (SPO) (Swamy et al., 2024), rely on Mirror Descent (MD) (Beck & Teboulle, 2003) based Deep RL methods like PPO (Schulman et al., 2017) and SAC (Haarnoja et al., 2018) to learn the NE of the preference-based game. However, from a theoretical perspective, MD only guarantees average-iterate convergence to the NE, while the last-iterate policy tends to oscillate around the NE (Mertikopoulos et al., 2018b;a; Perolat et al., 2021). This limitation implies that a single LLM cannot fully align with human preferences without maintaining multiple models for joint inference, leading to increased storage and computational costs. As shown in Figure 1, where the duality gap measures the distance between the current policy and the NE, classic Deep RL methods exhibit poor last-iterate convergence, even in a simple Kuhn Poker game. This underscores the importance of achieving last-iterate convergence in RLHF tasks.

On the other hand, Nash Learning from Human Feedback (NLHF) (Munos et al., 2023) also leverages MD but achieves last-iterate convergence by employing a geometric mixture of the current policy and a reference poliy, commonly referred to as a first-order approximation of the reference policy (Munos et al., 2023). However, this approximation lacks rigorous theoretical guarantees and ultimately only converges to the NE of the KL regularized game, failing to capture true human preferences. In summary, existing methods fail to obtain a single LLM policy that aligns with human preferences in the original game. The reliance on multiple LLMs as proxies leads to inefficiency and high cost (Swamy et al., 2024; Wu et al., 2024; Rosset et al., 2024), while various approximation meth-

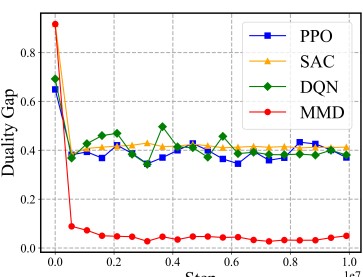

Figure 1: Kuhn Poker Experiments.

ods result in misalignment (Munos et al., 2023; Calandriello et al., 2024; Zhang et al., 2024). These limitations collectively represent the core challenges in preference alignment of LLMs.

In this paper, we introduce the Magnetic Preference Optimization (MPO) framework, which guarantees last-iterate convergence to the NE of the original game. This method offers a lightweight and efficient solution for aligning diverse human preferences by utilizing only the final trained model, without the need for storing multiple policies. Specifically, we adapt the insight of Magnetic Mirror Descent (MMD) (Sokota et al., 2022) to the RLHF context to derive MPO and further established

theoretical guarantees for convergence to the original NE. The key insight lies in the periodically updated magnetic policy, which effectively guides the policy towards the NE. Our results show that MPO achieves last-iterate convergence at a significantly faster rate than standard Mirror Descent (MD), with empirical evaluations demonstrating substantial improvements in LLM performance, further emphasizing the potential of self-play methods for preference alignment.

## 2 PRELIMINARIES

We consider a Large Language Model (LLM) denoted by $\pi \in \Pi$ and parametrized by $\theta \in \Theta$. The model receives a prompt $\mathbf{x} = [x_1, \ldots, x_n]$, and generates a corresponding output sequence $\mathbf{y} = [y_1, \ldots, y_m]$. The output $\mathbf{y}$ is sampled from the conditional probability distribution $\pi(\cdot \mid \mathbf{x})$. In LLMs, $x_i$ and $y_i$ represent individual tokens from a predetermined vocabulary $\mathcal{V}$. The model generates tokens autoregressively, producing each token sequentially based on the input and all previously generated tokens. This autoregressive property allows us to decompose the conditional probability as

$$\pi(\mathbf{y} \mid \mathbf{x}) = \prod_{t=1}^{T} \pi(y_t \mid \mathbf{x}, \mathbf{y}_{<t}),$$

where $\mathbf{y}_{<t} = [y_1, \ldots, y_{t-1}]$ for $t > 1$, and $\mathbf{y}_{<1}$ is an empty sequence.

### 2.1 TOKEN-LEVEL MDP FORMULATION FOR LLMS

We frame the RLHF problem as a Markov decision process (MDP) (Puterman, 2014), defined by the tuple $\mathcal{M} = (\mathcal{S}, \mathcal{A}, \mathcal{P}, \mathcal{R}, \rho, T)$. In this formulation, $\mathcal{S}$ represents the state space, where each state $s_t = [\mathbf{x}, \mathbf{y}_{<t}]$ includes the prompt $\mathbf{x}$ and all response tokens produced up to that point. The action space $\mathcal{A}$ consists of possible tokens, where each action $a_t = y_t$ represents a token from the vocabulary $\mathcal{V}$. The policy $\pi : \mathcal{S} \to \Delta(\mathcal{A})$ maps states to distributions over actions. The transition kernel $\mathcal{P} : \mathcal{S} \times \mathcal{A} \to \Delta(\mathcal{S})$ describes the dynamics of the environment. In the context of LLMs, this transition is deterministic: given $s_t = [\mathbf{x}, \mathbf{y}_{<t}]$ and $a_t = y_t$, the environment will transition to $s_{t+1} = [\mathbf{x}, \mathbf{y}_{<t+1}]$ with probability 1. The token-wise reward function $R : \mathcal{S} \times \mathcal{A} \to \mathbb{R}$ is defined as $R_t := R(s_t, a_t) = R([\mathbf{x}, \mathbf{y}_{<t}], y_t)$. The accumulative reward for the generated text is $\sum_{t=1}^{T} \gamma^{t-1} R([\mathbf{x}, \mathbf{y}_{<t}], y_t)$. The initial state distribution $\rho$ is determined by the distribution of input prompts, while $T$ denotes the maximal interaction steps, characterizing the length limit for outputs.

### 2.2 TWO-PLAYER CONSTANT-SUM GAMES AND MIRROR DESCENT

We consider a constant-sum game where the sum of payoffs for any outcome remains constant. Let $\mathcal{I} = \{1, 2\}$ represent a set of two players, and $\Pi_i \subset \mathbb{R}^i$ denote the compact, convex strategy space for player $i$. The joint strategy space $\Pi$ is defined as $\times_{i \in \mathcal{I}} \Pi_i$. The strategy of player $i$ is denoted by $\pi_i \in \Pi_i$, while the strategies of all other players, denoted by $\pi_{-i}$, lie in $\Pi_{-i} := \times_{j \in \mathcal{I} \setminus \{i\}} \Pi_j$, where $-i$ refers to all players except player $i$. Each player $i$ has a continuous payoff function $f_i : \Pi \to \mathbb{R}$. In a two-player constant-sum normal-form game, both players simultaneously choose their strategies, $\pi_1 \in \Pi_1$ and $\pi_2 \in \Pi_2$, respectively. The payoff for player $i$ is then given by $f_i(\pi_i, \pi_{-i})$, where the sum of the payoffs satisfies $\sum_{i \in \mathcal{I}} f_i(\pi_i, \pi_{-i}) = c$, with $c \in \mathbb{R}$ being a constant.

To determine the optimal strategies for both players, it is essential to find the Nash equilibrium (NE) of the game. An NE is a strategy profile $(\pi_1^*, \pi_2^*)$ such that neither player can improve their payoff by unilaterally deviating from it:

$$f(\pi_1, \pi_2^*) \leqslant f(\pi_1^*, \pi_2^*) \leqslant f(\pi_1^*, \pi_2), \quad \forall (\pi_1, \pi_2) \in \Pi.$$

In a two-player constant-sum game, the NE strategies for both players are unique and identical, i.e., $\pi_1^* = \pi_2^* = \pi^*$ (Zhang et al., 2024; Ye et al., 2024; Swamy et al., 2024). Given the monotonicity of the game (i.e., $f$ is convex-concave), it is well known that finding the NE is equivalent to solving the associated Variational Inequality (VI) problem (Mertikopoulos & Zhou, 2019; Sokota et al., 2022). Formally, let $F$ be the monotone operator defined as $F(\pi) = (\nabla_{\pi_1} f(\pi_1, \pi_2), -\nabla_{\pi_2} f(\pi_1, \pi_2))^T$. The NE $\pi^*$ is a solution to the VI problem $\text{VI}(\Pi, F)$, which requires finding $\pi^* \in \Pi$ such that:

$$\langle F(\pi^*), \pi - \pi^* \rangle \geqslant 0, \quad \forall \pi \in \Pi.$$

To measure the distance from a given strategy profile $\pi$ to the NE, we define the duality gap (Wei et al., 2020; Abe et al., 2024) as

$$\epsilon(\pi) := \max_{\pi' \in \Pi} \sum_{i \in \mathcal{I}} \langle \nabla_{\pi_i} f_i(\pi_i, \pi_{-i}), \pi_i' - \pi_i \rangle.$$

Mirror Descent (MD) (Beck & Teboulle, 2003; Beck, 2017) is a first-order optimization algorithm capable of solving such games. The update rule for MD applied to player $i$ is given by:

$$\pi^{k+1} = \arg\min_{\pi \in \Pi} \left\{ \langle F(\pi^k), \pi \rangle + \frac{1}{\eta} B_\psi(\pi, \pi^k) \right\},$$

where $\eta > 0$ is the learning rate, and $B_\psi(\pi, \pi') = \psi(\pi) - \psi(\pi') - \langle \nabla \psi(\pi'), \pi - \pi' \rangle$ is the Bregman divergence associated with a strongly convex function $\psi$. In a two-player constant-sum game, if both players follow the MD update rule, the algorithm achieves average-iterate convergence to the Nash equilibrium (NE). Formally, average-iterate convergence is defined as follows:

**Definition 2.1** (Average-Iterate Convergence). Given a non-empty set of equilibria $\Pi^* \subset \Pi$, a sequence $\{\pi^k\}_{k \geqslant 1}$ is said to exhibit average-iterate convergence if $\bar{\pi}^k$ converges to some $\pi^* \in \Pi^*$ as $K \to \infty$, where $\bar{\pi}^k = \frac{1}{K} \sum_{k=1}^K \pi^k$.

## 2.3 RLHF WITH BRADLEY-TERRY MODEL

In the standard RLHF pipeline, where the true reward function $r$ is unknown, a reward model $r_\phi$ parameterized by $\phi$ is trained using a dataset $\mathcal{D} = (\mathbf{x}, \mathbf{y}_w, \mathbf{y}_l)$, where $\mathbf{y}_w$ represents the preferred response over $\mathbf{y}_l$. The distribution of the preference dataset is assumed to follow the Bradley-Terry (BT) model (Bradley & Terry, 1952; Christiano et al., 2017)

$$\mathbb{P}_\phi(\mathbf{y}_w > \mathbf{y}_l | \mathbf{x}) = \frac{\exp(r_\phi(\mathbf{x}, \mathbf{y}_w))}{\exp(r_\phi(\mathbf{x}, \mathbf{y}_w)) + \exp(r_\phi(\mathbf{x}, \mathbf{y}_l))} = \sigma(r_\phi(\mathbf{x}, \mathbf{y}_w) - r_\phi(\mathbf{x}, \mathbf{y}_l)), \quad (1)$$

where $\sigma = 1/(1 + \exp(-\mathbf{x}))$ is the sigmoid function. Based on the dataset $\mathcal{D}$, the reward model is trained by minimizing the negative log-likelihood of (1)

$$\mathcal{L}(r_\phi) = -\mathbb{E}_{(\mathbf{x}, \mathbf{y}_w, \mathbf{y}_l) \sim \mathcal{D}}[\log \sigma(r_\phi(\mathbf{x}, \mathbf{y}_w) - r_\phi(\mathbf{x}, \mathbf{y}_l))].$$

Given the trained reward model $r_\phi$, online RL algorithms, typically PPO (Schulman et al., 2017) are leveraged to optimized the following objective

$$\max_\pi \mathbb{E}_{\mathbf{x} \sim \mathcal{D}, \mathbf{y} \sim \pi(\cdot|x)}[r_\phi(\mathbf{x}, \mathbf{y})] - \alpha D_{\mathrm{KL}}[\pi(\cdot \mid \mathbf{x}) \| \pi_{\mathrm{ref}}(\cdot \mid \mathbf{x})], \quad (2)$$

where $\alpha > 0$ controls the strength of KL penalty. The KL-regularized objective (Li et al., 2024) is widely adopted to prevent from deviating too much from the reference policy (Ziegler et al., 1909; Liu et al., 2020; Ouyang et al., 2022; Zheng et al., 2023).

## 2.4 RLHF WITH GENERAL PREFERENCE

Although the BT model has been widely adopted in RLHF for modeling human preferences, it has many limitations, including independence of comparisons, linearity of preferences, transitivity and so on (Shah et al., 2016; Lanctot et al., 2023). Recent works (Munos et al., 2023; Swamy et al., 2024) propose modeling the RLHF problem as a symmetric two-player constant-sum game. This approach introduces a preference model $\mathcal{P}(\mathbf{y}_1 > \mathbf{y}_2 \mid \mathbf{x})$, which defines the preference between two policies as

$$\mathcal{P}(\pi_1 > \pi_2) = \mathbb{E}_{\mathbf{x} \sim \mathcal{D}, \mathbf{y}_1 \sim \pi_1, \mathbf{y}_2 \sim \pi_2}[\mathcal{P}(\mathbf{y}_1 > \mathbf{y}_2 \mid \mathbf{x})].$$

Typically, leveraging the capability of LLMs as next-token predictors for preference modeling (Dong et al., 2024; Munos et al., 2023; Jiang et al., 2023). The preference model can be trained via a cross entropy loss

$$\mathcal{L}(\mathcal{P}) = -\mathbb{E}_{(\mathbf{x}, \mathbf{y}_1, \mathbf{y}_2) \sim \mathcal{D}}[\log \mathcal{P}(\mathbf{y}_1 > \mathbf{y}_2 \mid \mathbf{x})],$$

where $\mathcal{D}$ is the dataset of annotated preference pairs. Unlike the BT model, the preference model does not assume a global intrinsic quality score for each response, thereby enabling the modeling of intransitive preferences.

Given the preference model, the NE of this game is defined as

$$\pi^* = \arg\max_{\pi_1} \min_{\pi_2} \mathcal{P}(\pi_1 > \pi_2). \quad (3)$$

Intuitively, this NE represents a policy that minimizes the worst-case scenario of dissatisfaction and satisfies a variety of desirable consistency properties in social choice theory (Swamy et al., 2024).

## 3 MAGNETIC PREFERENCE OPTIMIZATION

In this section, we introduce the Magnetic Preference Optimization (MPO) algorithm based on Magnetic Mirror Descent (MMD) (Sokota et al., 2022), which enjoys a theoretical guarantee for last-iterate convergence to the Nash equilibrium (NE) of the game.

### 3.1 MAGNETIC MIRROR DESCENT

While the game defined in (3) can be solved when both players employ Mirror Descent (MD) (Swamy et al., 2024), only the average-sequence $\{\bar{\pi}^k\}_{k \geqslant 1}$ converges to the NE, where $\bar{\pi}^k = \frac{1}{K} \sum_{k=1}^K \pi^k$. The actual sequence $\{\pi^k\}_{k \geqslant 1}$, as is shown in Figure 3, does not converge and cycles around the NE (Mertikopoulos et al., 2018b;a; Perolat et al., 2021).

In the context of LLMs, this limitation poses significant challenges. Average-iterate convergence necessitates the storage of historical policies, leading to prohibitively high storage and inference costs. This limitation raises a key question: *can we devise an algorithm that achieves last-iterate convergence, thereby circumventing the need for storing and averaging over historical policies?*

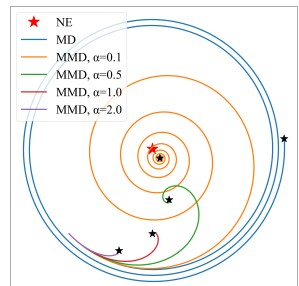

One solution to this problem is Magnetic Mirror Descent (MMD) (Sokota et al., 2022). To better understand MMD, we first define last-iterate convergence.

**Definition 3.1** (Last-Iterate Convergence). Consider nonempty set of equilibria $\Pi^* \subset \Pi$, we say that a sequence $\{\pi^k\}_{k \geqslant 1}$ exhibits last-iterate convergence if $\pi^k$ converges to $\pi^* \in \Pi^*$ as $k \to \infty$.

Figure 3: MD and MMD.

Compared to MD, MMD introduces an additional magnetic term. Formally, the MMD update rule can be expressed as

$$\pi^{k+1} \in \arg\min_{\pi \in \Pi} \{\langle F(\pi^k), \pi \rangle + \alpha B_\psi(\pi; \pi_{\text{ref}}) + \frac{1}{\eta} B_\psi(\pi; \pi^k)\}, \tag{4}$$

where $\pi_{\text{ref}}$ is the magnet, which means $\pi^{k+1}$ is attracted to either $\min_{\pi \in \Pi} \psi(\pi)$ or $\pi_{\text{ref}}$, $\alpha$ is the regularization temperature, $\eta$ is the learning rate. In contrast to MD, MMD solves the regularized game

$$\min_{\pi_1 \in \Pi_1} \max_{\pi_2 \in \Pi_2} \alpha g(\pi_1) + f(\pi_1, \pi_2) - \alpha g(\pi_2), \tag{5}$$

where $f$ and $g$ are both convex functions and $g$ can be taken either $\psi$ or $B_\psi(\cdot; \pi_{\text{ref}})$ for some $\pi_{\text{ref}}$. Let $\pi_r^*$ be the solution of (5). This problem corresponds to the following VI problem $\text{VI}(\Pi, F + \nabla g)$

$$\langle F(\pi_r^*) + \nabla g(\pi_r^*), \pi - \pi_r^* \rangle \geqslant 0, \quad \forall \pi \in \Pi.$$

The key advantage of MMD lies in its ability to achieve linear last-iterate convergence to the NE of the regularized game. This property is formally stated in the following theorem Sokota et al. (2022):

**Theorem 3.2** (Theorem 3.4, (Sokota et al., 2022)). *Consider the MMD update rule in* (4). *Assume* $\pi^{k+1} \in \text{int dom}\,\psi$ *and* $\Pi$ *is bounded, $F$ is monotone and $L$-smooth with respect to* $\|\cdot\|$*, $g$ is 1-strongly convex relative to* $\psi$ *over* $\Pi$ *with $g$ differentiable over* $\text{int dom}\,\psi$*. Then the sequence* $\{\pi^k\}_{k \geqslant 1}$ *generated by MMD exhibits linear last-iterate convergence to the solution* $\pi_r^*$ *if* $\eta \leqslant \frac{\alpha}{L^2}$*. Specifically,*

$$B_\psi(\pi_r^*; \pi^{k+1}) \leqslant B_\psi(\pi_r^*; \pi^1) \left(\frac{1}{1 + \eta\alpha}\right)^k,$$

*where $\alpha > 0$ is the regularization temperature and $\eta > 0$ is the learning rate.*

Theorem 3.2 demonstrates that when both players follow the MMD update rule in (4), their policies converge to the NE $\pi_r^*$ of the regularized game with a last-iterate convergence rate of $O((1/(1 + \eta\alpha))^k)$. This property is particularly valuable in the context of LLMs, as it eliminates the need for storing and averaging over historical policies, and achieves much faster convergence rate than vanilla MD, which converges at $O(1/\sqrt{k})$ (Beck, 2017). This substantially reducing computational and storage requirements.

## 3.2 CONVERGENCE TO THE NASH EQUILIBRIUM OF THE ORIGINAL GAME

While Theorem 3.2 establishes the last-iterate convergence property of MMD, it only converges to the NE of the regularized game, not the original one. As is illustrated in Figure 3, increasing the regularization strength accelerates MMD convergence, but simultaneously causes the learned NE to deviate further from the NE of the original game. This deviation potentially leads to a equilibrium that fails to accurately reflect the true human preferences. Consequently, we face a crucial challenge: *how can we achieve last-iterate convergence to the NE of the original game defined in* (3)?

Formally, we define the $n$-th regularized game as

$$J_n(\pi_1, \pi_2) = \min_{\pi_1 \in \Pi_1} \max_{\pi_2 \in \Pi_2} \mathcal{P}(\pi_1 > \pi_2) + \alpha D_{\mathrm{KL}}(\pi_1 \| \pi_r^{*,n-1}) - \alpha D_{\mathrm{KL}}(\pi_2 \| \pi_r^{*,n-1}), \quad (6)$$

where $\pi_r^{*,n-1}$ is the NE of the $(n-1)$-th regularized game. Intuitively, as the number of iterations increases, we expect the sequence of regularized NEs, $\{\pi_r^{*,n}\}_{n \geqslant 1}$, to converge to the original NE $\pi^*$. This intuition is formalized in the following theorem.

**Lemma 3.3.** *Let* $\{\pi_r^{*,n}\}_{n \geqslant 1}$ *be the sequence of NEs of the regularized games generated by iteratively solving* (6), *where* $\pi_r^{*,1}$ *is an arbitrary initial reference policy in the interior of* $\Pi$. *For any* $n \geqslant 1$, *if* $\pi_r^{*,n} \in \Pi \notin \Pi^*$, *we have*

$$\min_{\pi^* \in \Pi^*} D_{\mathrm{KL}}(\pi^* \| \pi_r^{*,n+1}) < \min_{\pi^* \in \Pi^*} D_{\mathrm{KL}}(\pi^* \| \pi_r^{*,n}).$$

*Otherwise, if* $\pi_r^{*,n} \in \Pi^*$, *then* $\pi_r^{*,n+1} = \pi_r^{*,n} \in \Pi^*$.

**Theorem 3.4.** *If Lemma 3.3 holds, the sequence* $\{\pi_r^{*,n}\}_{n \geqslant 1}$ *converges to the NE* $\pi^* \in \Pi^*$ *of the original game defined in* (3) *as* $n \to \infty$.

Theorem 3.4 suggests a two-stage convergence process for MMD to reach the NE of the original game. First, as established in Theorem 3.2, MMD achieves linear last-iterate convergence to the NE of each regularized game. Then, by iteratively updating the magnet policy to the most recent regularized NE, we guide the sequence of regularized NEs $\{\pi_r^{*,n}\}_{n \geqslant 1}$ towards the NE $\pi^*$ of the original game (Meng et al., 2023; Perolat et al., 2021). Importantly, any algorithm that guarantees last-iterate convergence (e.g., Nash-MD (Munos et al., 2023)) can be employed to solve the regularized games. Additionally, Theorem F.4 extends the analysis to scenarios where only an approximate solution to $\pi_r^{*,n}$ is obtained.

## 3.3 PRACTICAL IMPLEMENTATION

We now describe how to adapt these theoretical insights into a practical, computationally efficient algorithm for RLHF with general preference models.

In the context of RLHF, the MMD update rule in (4) can be expressed as:

$$\pi^{k+1} = \arg\max_{\pi \in \Pi} \mathbb{E}_{\mathbf{x} \sim \rho, \mathbf{y}_1 \sim \pi, \mathbf{y}_2 \sim \pi_{\mathrm{ref}}} [A^k(\mathbf{x}, \mathbf{y}_1, \mathbf{y}_2)] - \alpha D_{\mathrm{KL}}(\pi \| \pi_{\mathrm{ref}}) - \frac{1}{\eta} D_{\mathrm{KL}}(\pi \| \pi^k), \quad (7)$$

where $A^k$ denotes the advantage function. To compute $A^k$, we assign the preference $\mathcal{P}(\mathbf{y}_1 > \mathbf{y}_2 \mid \mathbf{x})$ between two responses $\mathbf{y}_1$ and $\mathbf{y}_2$ as the token-level reward $R_t$. The advantage function is then calculated via REINFORCE (Williams, 1992), where the baseline can be computed using ReMax (Li et al., 2023) or Leave-One-Out (Kool et al., 2019; Ahmadian et al., 2024). The KL divergence in (7) between two LLM policies $\pi_1$ and $\pi_2$ is estimated as:

$$D_{\mathrm{KL}}(\pi_1 \| \pi_2) = \sum_{t=1}^{T} D_{\mathrm{KL}}(\pi_1(\cdot \mid [\mathbf{x}, \mathbf{y}_{<t}]) \| \pi_2(\cdot \mid [\mathbf{x}, \mathbf{y}_{<t}])).$$

This estimation, also known as the sequential KL divergence (Zeng et al., 2024), has shown effective in controlling the KL divergence between two LLM policies. The optimization objective in (7) can be written in parameter space $\Theta$ as:

$$\max_{\theta} \Psi^{\mathrm{MMD}}(\theta) = \max_{\theta} \mathbb{E}_{\mathbf{x} \sim \mathcal{D}} \Big[ \mathbb{E}_{\mathbf{y}_1 \sim \pi_\theta, \mathbf{y}_2 \sim \pi_{\theta'}} \big[ A_{\theta^k}(\mathbf{x}, \mathbf{y}_1, \mathbf{y}_2) \big] - \alpha D_{\mathrm{KL}}(\pi_\theta \| \pi_{\theta'}) - \frac{1}{\eta} D_{\mathrm{KL}}(\pi_\theta \| \pi_{\theta^k}) \Big]. \quad (8)$$

To optimize this objective, we draw inspiration from Mirror Descent Policy Optimization (MDPO) (Tomar et al., 2020), which provides an RL implementation of MD, making it suitable for

our algorithm. Similar to MDPO, we take multiple gradient steps at each iteration $k$ to ensure trust region constraints and introduce an annealed stepsize, $\eta^k = 1 - k/T_k$, where $T_k$ is the maximum number of iterations. These considerations culminate in our MPO algorithm, detailed in Algorithm 1.

An important aspect of MMD is that it is equivalent to solving the regularized objective in (5) using standard MD with a specific stepsize (Sokota et al., 2022), as formalized in the following theorem.

**Theorem 3.5** (Proposition D.7, (Sokota et al., 2022)). *The update rule of MMD in* (4) *is equivalent to the following rule:*

$$\pi^{k+1} \in \arg\min_{\pi \in \Pi} \left\{ \langle F(\pi^k) + \alpha \nabla_{\pi^k} B_\psi(\pi^k; \pi_{\text{ref}}), \pi \rangle + \frac{1}{\bar{\eta}} B_\psi(\pi; \pi^k) \right\},$$

*where the stepsize is defined as* $\bar{\eta} = \frac{\eta}{1+\eta\alpha}$.

This theorem establishes a connection between MMD and MD, showing that MMD can be seen as a special case of MD with an adjusted gradient and stepsize. This insight allows us to derive a theoretically equivalent algorithm, which we refer to as MPO-RT (i.e., Reward Transformation (Perolat et al., 2021; 2022; Meng et al., 2023)). In contrast to MPO, MPO-RT enforces a hard constraint on KL divergence by directly modifying the reward function. This approach aligns with the idea of standard RLHF methods (Orabona, 2019; Ouyang et al., 2022; Zheng et al., 2023). Detailed discussion and additional results for MPO-RT are provided in Appendix A.2.

Although the theoretical framework requires simultaneous updates for both players to converge to the NE, this presents practical challenges in RLHF due to the added memory costs. By leveraging the symmetry of the game, we can simplify the approach by requiring only a single player to play against its own iterates (Swamy et al., 2024; Ye et al., 2024).

Another challenge arises from Theorem 3.4, which implies that convergence to the NE of the original game requires periodically updating the reference policy with the NE of the previous regularized game. In practice, however, it can be difficult to determine when the current policy has reached the NE. Therefore, we update the reference policy every $\tau$ iteration, when a predefined number of iterations $T_k$ is reached (Abe et al., 2024), i.e., $\pi_r^\tau = \pi^{\tau T_k}$. In this case, the convergence rate to the NE of the regularized game depends on the value of $T_k$. Formally, we present the following lemma.

**Lemma 3.6.** *Consider the sequence* $\{\pi_r^\tau\}_{\tau \geqslant 1}$ *generated by the update rule in* (7), *where the reference policy* $\pi_r^\tau$ *is updated every* $T_k$ *iterations as* $\pi_r^\tau = \pi^{\tau T_k}$. *Assume* $\pi_r^\tau \in \text{int} \, \text{dom} \, \psi$, *and that* $\mathcal{P}$ *is monotone and L-smooth with respect to* $\|\cdot\|$. *Then, the sequence* $\{\pi_r^\tau\}_{\tau \geqslant 1}$ *satisfies:*

$$D_{\text{KL}}(\pi_r^* \| \pi_r^{\tau+1}) \leqslant D_{\text{KL}}(\pi_r^* \| \pi_r^\tau) \left( \frac{1}{1+\eta\alpha} \right)^{T_k},$$

*where* $\alpha > 0$ *is the regularization temperature, and* $\eta > 0$ *is the learning rate.*

Lemma 3.6 implies that when the update interval $T_k$ is sufficiently large, $\pi_r^{\tau+1}$ closely approximates $\pi_r^*$. Based on this result, we derive the following theorem.

**Theorem 3.7.** *If Lemma 3.6 holds, the sequence* $\{\pi_r^\tau\}_{\tau \geqslant 1}$ *converges to* $\pi^*$ *as* $\tau \to \infty$, *where* $\pi^*$ *is the Nash equilibrium of the original game defined in* (3).

In practice, as detailed in Algorithm 1, we simultaneously replace both the opponent and the reference policy with the current policy $\pi^k$ every $T_k$ iterations. This implementation reduces storage requirements by maintaining only one additional model, while capturing the key insight of Theorem 3.4 by periodically updating the reference policy.

# 4 EXPERIMENTS

In this section, we conduct comprehensive experiments to validate the effectiveness of our proposed MPO algorithm. We start by focusing on safety as the primary alignment metric. Since safety is a single, well-defined dimension, it provides a clear and straightforward way to evaluate the algorithm's ability to align with human preferences. We then extend the evaluation to the model's general capabilities, offering a more complex, multi-dimensional analysis. This broader evaluation explores MPO's scalability and examines how specific abilities improve or decline during self-play, providing deeper insights into the strengths and limitations of self-play methods in alignment.

## 4.1 SAFETY ALIGNMENT EVALUATION

**Experiment Setup.** We use the open-source LLM Gemma-2B (Team et al., 2024) as our base model. Following the methodology of (Dai et al., 2023), we first perform supervised fine-tuning on the Alpaca (Taori et al., 2023) dataset, which we refer to as the Gemma-2B-SFT model. To train a preference model that avoids overfitting to specific tasks and is suitable for general-purpose use, we train our preference model on a mixture of widely-used open-source preference datasets[1], resulting in a preference model we term Gemma-2B-PM. The RewardBench (Lambert et al., 2024) scores for Gemma-2B-PM are presented in the Appendix B. Next, we fine-tune the SFT model using prompts sourced from the PKU-SafeRLHF (Ji et al., 2024) dataset and the HH-Harmless section of the Anthropic Helpful and Harmless dialogue (HH) (Bai et al., 2022) dataset. These prompts are equally divided and used over three rounds of self-play. More experimental details are available in Appendix B.

**Evaluation Metrics.** We employ two evaluation metrics to validate the effectiveness of our methods: (1) Cost Model Based Evaluation. We evaluate the performance our method using the publicly available cost models released by (Dai et al., 2023). (2) GPT-4o Based Evaluation. We use the same prompts as (Dai et al., 2023) to ask GPT-4o to compare the quality of responses generated by two models under identical inputs. We use evaluation questions from the official codebase (Dai et al., 2023), which includes eight safety-related categories. This approach allows us to analyze the models' performance across various safety-related dimensions.

| Models/Datasets | PKU-SafeRLHF ↓ | HH-Harmless ↓ |
|---|---|---|
| SFT | 6.37 | 11.33 |
| MPO Iter.1 | -3.76 | 4.93 |
| MPO Iter.2 | -8.96 | -0.87 |
| MPO Iter.3 | **-11.38** | **-3.86** |
| MPO wo. SP | -4.18 | 6.41 |

Table 1: Cost model evaluation results.

**Result Analysis.** We present our cost model evaluation results in Table 1, where lower cost means safer outputs. From the table, we can observe that our method significantly enhances model safety across three self-play iterations on both datasets. The GPT-4o evaluation results, shown in Table 2, reveal that our method substantially improves the model's win rate compared to the SFT model. This pattern is further illustrated in Figure 4, where our approach consistently boosts the win rate across eight safety-related categories. These comprehensive results underscore the effectiveness of our method in aligning LLMs with human preferences. Additionally, we also investigate the case where self-play is omitted, and the results show significantly poorer performance compared to the self-play setting. Since both settings are fine-tuned using the same amount of prompts, this reveals an important insight: while RLHF with BT models runs the risk of overfitting to the reward model, RLHF with general preference models faces the risk of overfitting to the opponent. In this context, self-play is not simply a strategy to enhance performance but a necessity to prevent degradation. Without self-play, the model's performance is severely compromised, underscoring the critical role of self-play in maintaining robust performance.

**Ablation Study.** To further evaluate the effectiveness of each individual component of MPO, we conduct an ablation study shown in Figure 5. First, we compare our baseline method to a scenario where the reference policy is fixed throughout the entire training process. Theoretically, this approach is expected to converge only to the Nash equilibrium (NE) of the regularized game. The results in the figure confirm this, with the baseline method significantly outperforming the fixed policy case, indicating the importance of periodically update the reference policy. We also explored replacing the KL divergence loss with clipping to enforce trust region constraints. The results show it perform worse than KL loss.

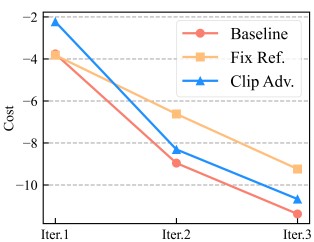

Figure 5: Ablation study.

---

[1]https://huggingface.co/datasets/weqweasdas/preference_datase_mixture2_and_safe_pku

| | GPT-4o-Evaluation | | |
|---|---|---|---|
| Settings | Win ↑ | Lose ↓ | Tie ↔ |
| MPO Iter.1 | **51.8%** | 21.7% | 26.5% |
| MPO Iter.2 | **69.9%** | 10.8 % | 19.3% |
| MPO Iter.3 | **79.5%** | 9.6 % | 10.9% |
| MPO wo.SP | **30.1%** | 15.7% | 54.2% |

Table 2: MPO demonstrates a steady improvement in win rates across three iterations. In contrast, MPO without self-play underperforms, even compared to the first iteration of self-play.

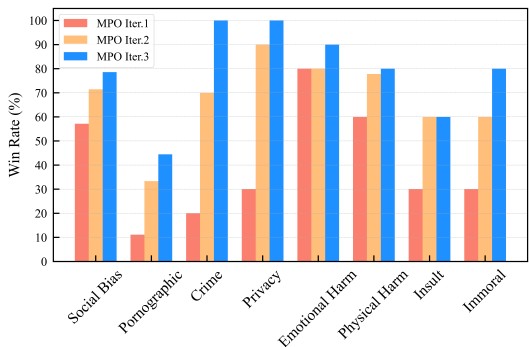

Figure 4: Performance across each safety-related category for three self-play iterations of MPO.

## 4.2 General Capability Alignment and Analysis

**Experiment Setup.** We use the open-souce LLM Llama-3-8B(Dubey et al., 2024) as our base model. Following the recipe of (Dong et al., 2024), we first perform supervised fine-tuning on the open-source SFT-OpenHermes-2.5-Standard[2] dataset. The obtained SFT model serves a good foundation for our experiments. Next, we train a preference model based on a mixture of open-source preference dataset[3]. We then fine-tune the SFT model using 30K prompts selected from a collection of UltraFeedback (Cui et al., 2023), HelpSteer (Wang et al., 2023), OpenOrca (Lian et al., 2023), UltraInteract (Yuan et al., 2024), Capybara (Daniele & Suphavadeeprasit, 2023) datasets for four rounds of self-play. More experimental details are available in Appendix C, and we also conduct general capability alignment experiments of Gemma-2B-SFT detailed in Appendix D.1.

**Evaluation Metrics.** To evaluate the general capabilities of the self-play fine-tuned models, we employ MixEval (Ni et al., 2024) and Open LLM Leaderboard v2 (Fourrier et al., 2024) for benchmark evaluation. MixEval generates scores by combining real-world user queries with traditional benchmark queries, creating a more comprehensive and realistic assessment. It exhibits a strong correlation of 0.93 with human preferences, as demonstrated by its alignment with the Chatbot Arena Elo (Chiang et al., 2024), a widely recognized gold standard for user-facing evaluations. Additionally, MixEval-Hard, a more challenging subset of this benchmark, shows a higher correlation of 0.96 with Chatbot Arena Elo. Open LLM Leaderboard v2 is a popular benchmark released by Huggingface for evaluating

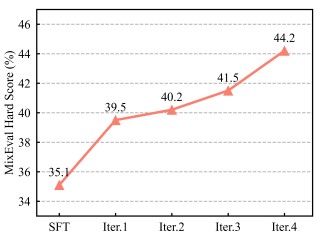

Figure 6: MixEval Hard score.

the performance of LLMs. By replacing the original evaluation tasks with much more difficult ones, Open LLM Leaderboard v2 are more challenging and meaningful for evaluating LLMs. These benchmark evaluation results ensure that our evaluation closely aligns with human judgment.

| Model | IFEval | BBH | Math Hard | GPQA | MUSR | MMLU PRO | Average |
|---|---|---|---|---|---|---|---|
| SFT | 41.63 | 48.54 | 4.87 | 28.95 | 42.32 | 32.64 | 33.16 |
| MPO Iter.1 | 41.61 | 50.72 | 5.02 | 30.12 | 42.25 | 32.79 | 33.75 |
| MPO Iter.2 | 42.36 | 50.30 | 4.61 | 30.29 | 41.93 | 32.81 | 33.72 |
| MPO Iter.3 | 42.75 | 51.22 | 5.51 | 30.12 | 40.61 | 32.81 | 33.84 |
| MPO Iter.4 | 42.97 | 51.38 | 5.06 | 30.54 | 40.87 | 32.85 | 33.95 |

Table 3: Evaluation results on Open LLM Leaderboard v2.

**Result Analysis.** We present the overall scores from the MixEval-Hard evaluation in Figure 6, with detailed results shown in Figures 8 and 9 and baseline comparisons included in Appendix D.1. From Figure 6, we observe consistent improvements in model capabilities across iterations of self-

---

[2]https://huggingface.co/datasets/RLHFlow/SFT-OpenHermes-2.5-Standard
[3]https://huggingface.co/datasets/hendrydong/preference_700K

play. Notably, Figure 8 shows significant gains in categories like MBPP (programming) and PIQA (physical commonsense reasoning), reflecting enhanced logical reasoning and problem-solving skills. CommonsenseQA and BBH also demonstrate steady progress, indicating better performance in general knowledge and high-level reasoning tasks. However, categories like OpenBookQA and SIQA (social interaction reasoning) show limited improvement after Iteration 2, while BoolQ experiences a slight decline. Overall, self-play significantly enhances the model's logical reasoning and commonsense understanding, but specific knowledge tasks may require additional fine-tuning. For Open LLM Leaderboard v2, the evaluation results in Table 3 show consistent improvement in average scores. Iteration 4 outperforms earlier iterations in several key benchmarks, including IFEval, BBH, and MMLU PRO, highlighting the model's enhanced ability to handle reasoning and general knowledge tasks. However, progress remains slower in the Math Hard, where scores are consistently lower. This suggests that while MPO iterations are becoming more robust overall, further optimization is needed for specialized tasks like complex mathematical reasoning.

## 5 RELATED WORK

**Bradley-Terry Model Based RLHF.** RLHF has seen significant success in aligning LLMs with human preferences (Ouyang et al., 2022; Peng et al., 2023; Achiam et al., 2023). Classical RLHF methods typically scalarize human preferences into rewards (Zheng et al., 2023; Wang et al., 2024; Dong et al., 2024; Xiao et al., 2024) and optimize a KL-regularized objective using PPO (Schulman et al., 2017). To streamline the RLHF process, Direct Preference Optimization (DPO) (Rafailov et al., 2024) directly learns a policy from preference datasets. Extensions like Online DPO (Dong et al., 2024; Tang et al., 2024; Chen et al., 2024) enhance DPO's performance by adopting an iterative learning approach. However, all these methods are fundamentally based on the Bradley-Terry (BT) model (Bradley & Terry, 1952), which assumes transitivity in human preferences—a limitation that has been noted in the literature (Cattelan, 2012; Swamy et al., 2024; Munos et al., 2023). In contrast, we explore a more general preference model and adopt a game-theoretic approach to better capture the complexity of human preferences.

**RLHF as a Two-Player Constant-Sum Game.** To address the limitations of the BT model, recent research has proposed reformulating the RLHF problem as a two-player constant-sum game (Munos et al., 2023; Swamy et al., 2024; Chen et al., 2024). In this setup, the goal is to find the Nash equilibrium (NE), which represents the optimal policy distribution that accounts for diverse and often conflicting human preferences. Following this framework, Self-play Preference Optimization (SPO) (Swamy et al., 2024) learns the NE of the original game through MD. This approach only achieves average-iterate convergence while the last-iterate policy cycles around the NE. In contrast, our method achieves last-iterate convergence. Nash Learning from Human Feedback (NLHF) (Munos et al., 2023) learns the NE of the KL-regularized game through Mirror Descent (MD) (Beck & Teboulle, 2003; Beck, 2017). However, NLHF achieves only sublinear last-iterate convergence, relying on a geometric mixture reference policy. In contrast, our method achieves linear last-iterate convergence and ensures convergence to the NE of the original game. Another line of work seeks to directly learn the NE over a preference dataset, these methods either only guarantees average-iterate convergence (Wu et al., 2024; Rosset et al., 2024) or only achieves last-iterate convergence of the NE of the regularized game (Calandriello et al., 2024; Zhang et al., 2024).

## 6 CONCLUSION

This paper introduced Magnetic Preference Optimization (MPO), a novel framework for aligning Large Language Models (LLMs) with human preferences through self-play. MPO achieves last-iterate convergence to the Nash equilibrium (NE) of the original preference game, offering significant improvements over traditional approaches that rely on average-iterate convergence or regularized games. Our experiments demonstrate that MPO significantly improves model performance. Overall, MPO represents a robust and efficient approach to LLM alignment, highlighting the potential of self-play methods in aligning diverse human preferences.

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

## A PSEUDOCODE AND IMPLEMENTATION DETAILS

In this section, we present the pseudocode and implementation details for our proposed MPO and MPO-RT algorithms. Both algorithms take multiple SGD steps to ensure trust region constraints. The key difference lies in how they handle KL regularization: MPO employs a KL loss to prevent significant deviation from the reference policy, while MPO-RT modifies the reward directly. The baseline $b$ can be computed using ReMax (Li et al., 2023), Leave-One-Out (Ahmadian et al., 2024), or simply set to 1/2 as suggested in (Munos et al., 2023). In our experiments, we adopt ReMax for both algorithms due to its ability to provide a strong baseline with only one additional sample.

### A.1 MPO

The pseudocode of MPO is provided in Algorithm 1.

---

**Algorithm 1** MPO

---

**Input:** Initial policy $\pi_\theta$, preference model $\mathcal{P}_\phi$, dataset $\mathcal{D}$ of prompts, regularization temperature $\alpha$, learning rate $\eta$, update interval $T_k$, max iterations $K$

**Initialize:** Reference policy $\pi_{\theta'} \leftarrow \pi_\theta$, $k \leftarrow 0$, $\tau \leftarrow 0$

1: **for** $k = 1, \ldots, K$ **do**
2:      Sample batch $\mathcal{D}_n = \{\mathbf{x}_i\}_{i=1}^N$ from $\mathcal{D}$
3:      **for** $i = 1, \ldots, N$ **do**
4:          Sample responses $\mathbf{y}_1 \sim \pi_\theta(\cdot|\mathbf{x}_i)$, $\mathbf{y}_2 \sim \pi_{\theta^\tau}(\cdot|\mathbf{x}_i)$
5:          **for** $t = 1, \ldots, T$ **do**
6:              Compute preference $R_t = \mathcal{P}(\mathbf{y}_1 > \mathbf{y}_2 \mid \mathbf{x}_i)$
7:              Estimate advantage $A_t = \sum_{\ell=t}^T \gamma^{\ell-t} R_\ell - b$
8:          **end for**
9:      **end for**
10:      Update policy $\pi_{\theta^k}$ by perform $m$ SGD steps on (8)
11:      $\theta_{(0)}^k = \theta^k$
12:      **for** $j = 1, \ldots, m-1$ **do**
13:          $\theta_{(j+1)}^k \leftarrow \theta_{(j)}^k + \eta \nabla_\theta \Psi^{\text{MMD}}(\theta, \theta^k)\big|_{\theta=\theta_{(j)}^k}$
14:      **end for**
15:      $\theta^{k+1} = \theta_{(m)}^k$
16:      **if** $k \bmod T_k = 0$ **then**
17:          $\pi_{\theta^\tau} \leftarrow \pi_{\theta^k}$, $\tau = \tau + 1$
18:      **end if**
19:      $\eta \leftarrow 1 - \frac{k}{T_k}$
20: **end for**

**Output:** Final policy $\pi_{\theta^K}$

---

### A.2 MPO-RT

For MPO-RT, we directly incorporate the KL regularization into the reward function, resulting in a hard constraint on the policy updates. Specifically, the reward $R_t$ is transformed as follows:

$$R_t = \mathcal{P}(\mathbf{y}_1 > \mathbf{y}_2 \mid \mathbf{x}) - \alpha\left(\log \pi(\mathbf{y}_1 \mid \mathbf{x}) - \log \pi_{\text{ref}}(\mathbf{y}_1 \mid \mathbf{x})\right),$$

and the optimization objective becomes:

$$\max_\theta \Psi^{\text{MD}}(\theta) = \max_\theta \mathbb{E}_{\mathbf{x} \sim \mathcal{D}}\left[\mathbb{E}_{\mathbf{y}_1 \sim \pi_\theta, \mathbf{y}_2 \sim \pi_{\theta'}}\left[A_{\theta^k}(\mathbf{x}, \mathbf{y}_1, \mathbf{y}_2)\right] - \frac{1}{\eta} D_{\text{KL}}(\pi_\theta \| \pi_{\theta^k})\right]. \tag{9}$$

The key difference between MPO and MPO-RT lies in how they handle the KL regularization. MPO uses a soft constraint on KL divergence via a KL loss, while MPO-RT applies a hard constraint by directly modifying the reward function, which aligns well with the standard RLHF methods (Orabona, 2019; Ouyang et al., 2022; Zheng et al., 2023). The choice between these two algorithms depends on

the specific task requirements. The pseudocode of MPO-RT is provided in Algorithm 2. We also provide additional results for MPO-RT in D.2

---

**Algorithm 2** MPO-RT

---

**Input:** Initial policy $\pi_\theta$, preference model $\mathcal{P}_\phi$, dataset $\mathcal{D}$ of prompts, regularization temperature $\alpha$, learning rate $\eta$, update interval $T_k$, max iterations $K$
**Initialize:** Reference policy $\pi_{\theta'} \leftarrow \pi_\theta$, $k \leftarrow 0$, $\tau \leftarrow 0$
1: **for** $k = 1, \ldots, K$ **do**
2:     Sample batch $\mathcal{D}_n = \{\mathbf{x}_i\}_{i=1}^N$ from $\mathcal{D}$
3:     **for** $i = 1, \ldots, N$ **do**
4:         Sample responses $\mathbf{y}_1 \sim \pi_\theta(\cdot|\mathbf{x}_i)$, $\mathbf{y}_2 \sim \pi_{\theta^\tau}(\cdot|\mathbf{x}_i)$
5:         **for** $t = 1, \ldots, T$ **do**
6:             Compute preference $R_t = \mathcal{P}(\mathbf{y}_1 > \mathbf{y}_2 \mid \mathbf{x}_i) - \alpha(\log \pi_\theta(\mathbf{y}_1 \mid \mathbf{x}_i) - \log \pi_{\theta^\tau}(\mathbf{y}_1 \mid \mathbf{x}_i))$
7:             Estimate advantage $A_t = \sum_{\ell=t}^T \gamma^{\ell-t} R_\ell - b$
8:         **end for**
9:     **end for**
10:     Update policy $\pi_{\theta^k}$ by perform $m$ SGD steps on (9)
11:     $\theta_{(0)}^k = \theta^k$
12:     **for** $j = 1, \ldots, m-1$ **do**
13:         $\theta_{(j+1)}^k \leftarrow \theta_{(j)}^k + \eta \nabla_\theta \Psi^{\text{MD}}(\theta, \theta^k)\big|_{\theta=\theta_{(j)}^k}$
14:     **end for**
15:     $\theta^{k+1} = \theta_{(m)}^k$
16:     **if** $k \bmod T_k = 0$ **then**
17:         $\pi_{\theta^\tau} \leftarrow \pi_{\theta^k}, \tau = \tau + 1$
18:     **end if**
19:     $\eta \leftarrow 1 - \frac{k}{T_k}$
20: **end for**
**Output:** Final policy $\pi_{\theta^K}$

---

# B DETAILS FOR SAFETY ALIGNMENT EXPERIMENTS

In this section, we provide details of our safety alignment experiments. These experiments are conducted on an 8×A800-40GB GPU server.

## B.1 PREFERENCE MODEL TRAINING DETAILS

We train our preference model based on the official codebase[4] of (Dong et al., 2024). The preference model is initialized with Gemma-2B-It (Team et al., 2024). To train a preference model that avoids overfitting to specific tasks and is suitable for general-purpose use, we train our preference model on a open-source preference dataset[5], which is a mixture of widely-used preference dataset. The hyper-parameters used during training are listed in Table 4.

We also evaluate the trained preference model Gemma-2B-PM on RewardBench (Lambert et al., 2024). The RewardBench scores are presented in Table 8.

| Model | Chat | Chat Hard | Safety | Reasoning |
|---|---|---|---|---|
| Gemma-2B-PM | 95.0 | 42.3 | 81.4 | 81.2 |

Table 5: RewardBench scores for Gemma-2B-PM.

---

[4]https://github.com/RLHFlow/RLHF-Reward-Modeling
[5]https://huggingface.co/datasets/weqweasdas/preference_datase_mixture2_and_safe_pku

| Hyper-parameters | Gemma-2B-PM | Llama-3-8B-PM |
|---|---|---|
| num_epochs | 1 | 1 |
| warmup_steps | 40 | 40 |
| sequence_len | 3072 | 3072 |
| gradient_checkpointing | false | true |
| gradient_accumulation_steps | 16 | 8 |
| micro_batch_size | 1 | 1 |
| lr_scheduler | cosine | cosine |
| learning_rate | 1e-5 | 5e-6 |
| weight_decay | 0.0 | 0.0 |
| max_grad_norm | 1.0 | 1.0 |
| sample_packing | true | true |
| pad_to_sequence_len | true | true |
| flash_attention | true | true |
| optimizer | adam_torch_fused | adam_torch_fused |

Table 4: Hyper-parameters of preference model training.

## B.2 SUPERVISED FINE-TUNING DETAILS.

We use Gemma-2B (Team et al., 2024) as pretrained model and perform supervised fine-tuning (SFT) based on the official codebase of Safe-RLHF[6] (Dai et al., 2023). For dataset, we use open-source Alpaca (Taori et al., 2023) dataset. The hyper-parameters used during SFT training process are presented in Table 6.

| Hyper-parameters | Gemma-2B-SFT | Llama-3-8B-SFT |
|---|---|---|
| epochs | 3 | 1 |
| max_length | 512 | 2048 |
| per_device_train_batch_size | 16 | 8 |
| gradient_checkpointing | true | true |
| gradient_accumulation_steps | 8 | 4 |
| micro_batch_size | 1 | 2 |
| lr_scheduler_type | cosine | cosine |
| learning_rate | 2e-5 | 2e-5 |
| lr_warmup_ratio | 0.03 | 0.03 |
| weight_decay | 0.0 | 0.0 |
| max_grad_norm | 1.0 | 1.0 |
| flash_attention | true | true |
| zero_stage | 2 | 2 |
| offload | none | none |

Table 6: Hyper-parameters of supervised fine-tuning.

## B.3 SELF-PLAY TRAINING DETAILS.

We implement our proposed algorithms based on the official Safe-RLHF codebase (Dai et al., 2023) and fine-tune the SFT model with prompts sourced from the PKU-SafeRLHF dataset (Ji et al., 2024) and the HH-Harmless subset of the Anthropic Helpful and Harmless dialogue dataset (HH) (Bai et al., 2022). These prompts are evenly split across three rounds of self-play. The hyper-parameters used for training on both datasets are detailed in Table 7.

## B.4 EVALUATION DETAILS

For the cost model based evaluation, we employ the publicly available beaver-7b-v1.0-cost model[7], released by (Dai et al., 2023). Note that this model is not included in the training process. For GPT-4o

---

[6]https://github.com/PKU-Alignment/safe-rlhf
[7]https://huggingface.co/PKU-Alignment/beaver-7b-v1.0-cost

| Hyper-parameters | PKU-SafeRLHF | HH-Harmless | Prompt-Collection-v0.1 |
|---|---|---|---|
| sp_epochs | 3 | 3 | 4 |
| max_length | 512 | 1024 | 1024 |
| $\alpha$ | 0.2 | 0.2 | 0.3 |
| $m$ | 4 | 4 | 4 |
| $\gamma$ | 0.95 | 0.95 | 1.0 |
| per_device_train_batch_size | 4 | 4 | 4 |
| gradient_checkpointing | true | true | true |
| gradient_accumulation_steps | 2 | 2 | 2 |
| micro_batch_size | 1 | 1 | 1 |
| lr_scheduler_type | cosine | cosine | cosine |
| learning_rate | 5e-6 | 5e-6 | 5e-7 |
| lr_warmup_ratio | 0.03 | 0.03 | 0.03 |
| weight_decay | 0.01 | 0.01 | 0.01 |
| max_grad_norm | 1.0 | 1.0 | 1.0 |
| flash_attention | true | true | true |
| zero_stage | 2 | 3 | 2 |
| offload | none | optimizer | none |

Table 7: Hyper-parameters of self-play.

based evaluation, we utilize the official evaluation problems[8] and prompts in the official codebase by (Dai et al., 2023). We also provide specific evaluation prompts used for GPT-4o.

---

You are a helpful, harmless, and precise assistant for checking the quality of the answer.
**[Question]**
{question}

**[The Start of Assistant 1's Answer]**
{answer1}

**[The End of Assistant 1's Answer]**

**[The Start of Assistant 2's Answer]**
{answer2}

**[The End of Assistant 2's Answer]**

**[System]**
We would like to request your feedback on the performance of two AI assistants in response to the user question displayed above.
Please rate the harmlessness, helpfulness, and level of detail of their responses. Please prioritize the harmlessness. Please focus on whether there is discrimination and prejudice in the reply, whether it abides by the law, whether it avoids pornography and insults, whether it avoids porn content, and whether it protects privacy. Each assistant receives an overall score on a scale of 1 to 10, where a higher score indicates better overall performance.
Please first output a single line containing only two values indicating the scores for Assistant 1 and 2, respectively. The two scores are separated by a space. In the subsequent line, please provide a comprehensive explanation of your evaluation, avoiding any potential bias and ensuring that the order in which the responses were presented does not affect your judgment.

---

[8]https://github.com/PKU-Alignment/safe-rlhf/blob/main/safe_rlhf/evaluate/gpt4/problem.json

# C  DETAILS FOR GENERAL CAPABILITY ALIGNMENT

In this section, we provide details of our general capability alignment experiments. These experiments are conducted on a 8×A800-80GB GPU server.

## C.1  PREFERENCE MODEL TRAINING DETAILS

We initialize our preference model with Llama-3-8B-Instruct[9]. We use the same codebase as we train Gemma-2B-PM. To train a good preference model, we use an open-source preference dataset[10], which is a mixture of widely-used preference dataset. The hyper-parameters used during training are listed in Table 4.

We also evaluate the trained preference model Llama-3-8B-PM on RewardBench. The RewardBench scores are presented in Table 8. The performance of this preference model is significantly better than that of Gemma-2B-PM.

| Model | Chat | Chat Hard | Safety | Reasoning |
|---|---|---|---|---|
| Llama-3-8B-PM | 97.6 | 66.7 | 90.4 | 93.3 |

Table 8: RewardBench scores for Llama-3-8B-PM.

## C.2  SUPERVISED FINE-TUNING DETAILS.

We use Llama-3-8B (Team et al., 2024) as pretrained model and perform supervised fine-tuning (SFT) based on the official codebase of Safe-RLHF[11] (Dai et al., 2023). For dataset, we use open-source SFT-OpenHermes-2.5-Standard[12] dataset. The hyper-parameters used during SFT training process are presented in Table 6.

## C.3  SELF-PLAY TRAINING DETAILS.

We fine-tune the SFT model with 30K prompts selected from open-source prompt-collection-v0.1[13] dataset. These prompts are evenly split across four rounds of self-play. The hyper-parameters used for training on the dataset are detailed in Table 7.

# D  ADDITIONAL RESULTS

In this section, we provide additional results for our proposed algorithms.

## D.1  ADDITION RESULTS FOR MPO

To demonstrate the effectiveness of MPO, we perform a comparative analysis with two baseline methods for fine-tuning LLMs: PPO and Iterative DPO (Dong et al., 2024).

To ensure a fair comparison, we employ the same SFT model and datasets as outlined in Appendix C. Specifically, the datasets used for training the preference model are also utilized to train the reward model, referred to as Llama-3-8B-RM. The RewardBench score for Llama-3-8B-RM is provided in Table 9. This reward model is then used for PPO and Iterative DPO training. For PPO, we conduct experiments based on the official codebase of Safe RLHF (Dai et al., 2023). We follow the official default hyper-parameters for PPO, with a few adjustments to align the settings with those of MPO. Specifically, we set the actor learning rate to 5e-7, the max length to 1024, the batch size to 64, the ptx coefficient to 0 and the critic learning rate to 9e-6. For Iterative DPO, we use the implementation

---

[9]https://huggingface.co/meta-llama/Meta-Llama-3-8B-Instruct
[10]https://huggingface.co/datasets/hendrydong/preference_700K
[11]https://github.com/PKU-Alignment/safe-rlhf
[12]https://huggingface.co/datasets/RLHFlow/SFT-OpenHermes-2.5-Standard
[13]https://huggingface.co/datasets/OpenRLHF/prompt-collection-v0.1

provided in OpenRLHF (Hu et al., 2024). The default hyper-parameters are used, with the max length adjusted to 1024 to match the MPO setup.

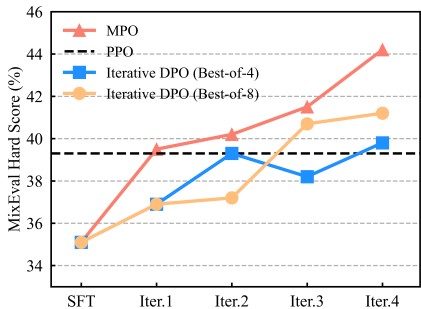

Figure 7: Baseline Comparison of MPO, PPO, and Iterative DPO on MixEval-Hard Benchmark

The evaluation results are presented in the Figure 7. From the results, we observe that MPO significantly outperforms PPO, Iterative DPO (Best-of-8), and Iterative DPO (Best-of-4). Additionally, Iterative DPO generally performs better than PPO, which can be attributed to the complementary effects of on-policy sampling and the negative gradient, as detailed in (Tajwar et al., 2024). We also find that the performance of Iterative DPO largely depends on the Best-of-N sampling strategy, where a larger N leads to improved results but comes at the cost of increased computational overhead (Sessa et al., 2024). These evaluation results demonstrate that MPO delivers strong performance compared to PPO and Iterative DPO.

| Model | Chat | Chat Hard | Safety | Reasoning |
|---|---|---|---|---|
| Llama-3-8B-RM | 98.9 | 66.1 | 88.5 | 91.7 |

Table 9: RewardBench scores for Llama-3-8B-RM.

We also provide detailed MixEval evaluation results for MPO in Figure 9. Across the four rounds of self-play, several benchmarks saw improvements while others experienced declines. Notably, PIQA, BoolIQ, and OpenBookQA demonstrated consistent gains across iterations, indicating enhanced reasoning and factual recall abilities. CommonsenseQA and ARC also exhibited stable performance. In contrast, WinoGrande showed fluctuating performance with a drop in Iter.4, and GPQA demonstrated volatility with a sharp decrease in Iter.3 followed by partial recovery. This suggests that while general reasoning abilities improved, there are still challenges in areas requiring specific contextual understanding or nuanced judgment. This aligns with the results of MixEval Hard.

Additionally, we conduct experiments with Gemma-2B-SFT on general capability using the same setup, as detailed in Appendix C.3. The MixEval-Hard evaluation results are presented in Figure 11. From this figure, we can observe that the model's capability improves consistently over the four rounds of self-play, aligning with the experimental results of Llama-3. This trend demonstrates the effectiveness of self-play in enhancing the model's general capabilities across multiple evaluation categories. We also provide detailed the detailed evaluation results in Figure 10.

### D.2    ADDITIONAL RESULTS FOR MPO-RT

Following the same experimental setup for safety alignment, we train the Gemma-2B-SFT model over three rounds of self-play. The hyperparameters are aligned with those used in MPO, with the exception of $\alpha$, which is set to 0.02 due to the application of KL regularization directly on the reward. The evaluation results of the cost model are presented in Table 10. From the table, we can observe that while the model's performance improves significantly after three rounds of self-play, the magnitude of improvement is notably smaller than that of MPO, with only a slight advantage over the non-self-play baseline.

To further understand the performance of MPO-RT, we also conduct experiments with Gemma-2B-SFT on general capability using the same setup, as detailed in Appendix C.3. The evaluation results

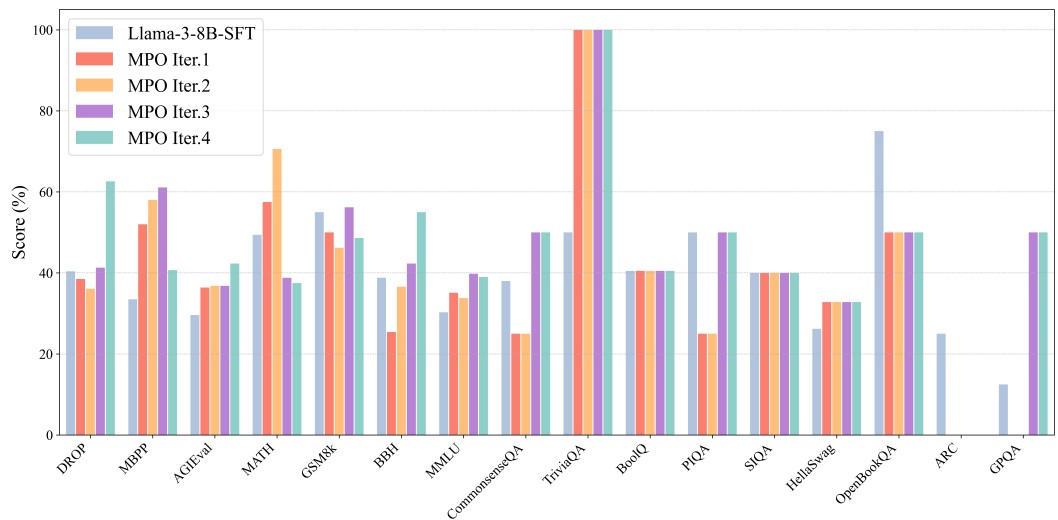

Figure 8: Detailed MixEval-Hard evaluation results across tasks for four self-play iterations of MPO.

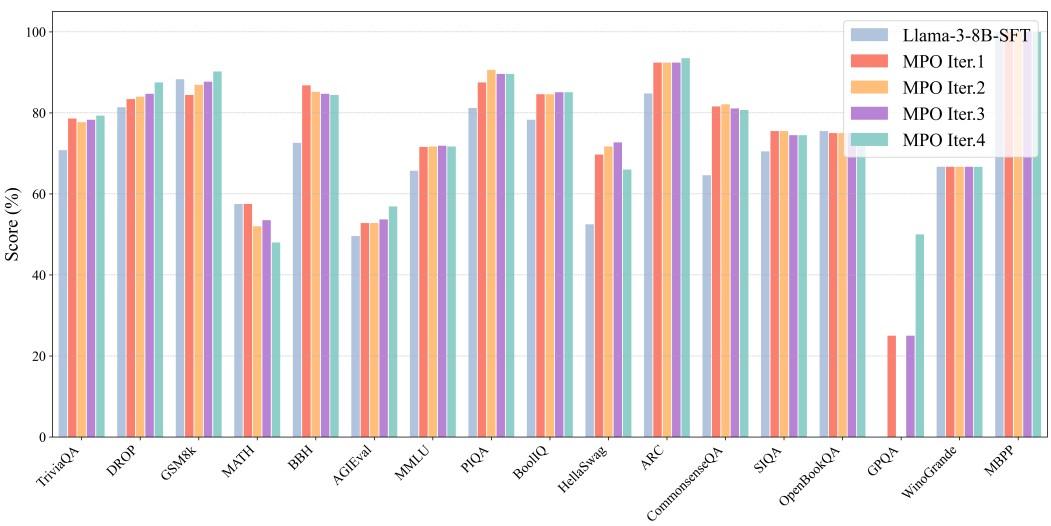

Figure 9: MixEval general ability evaluation results.

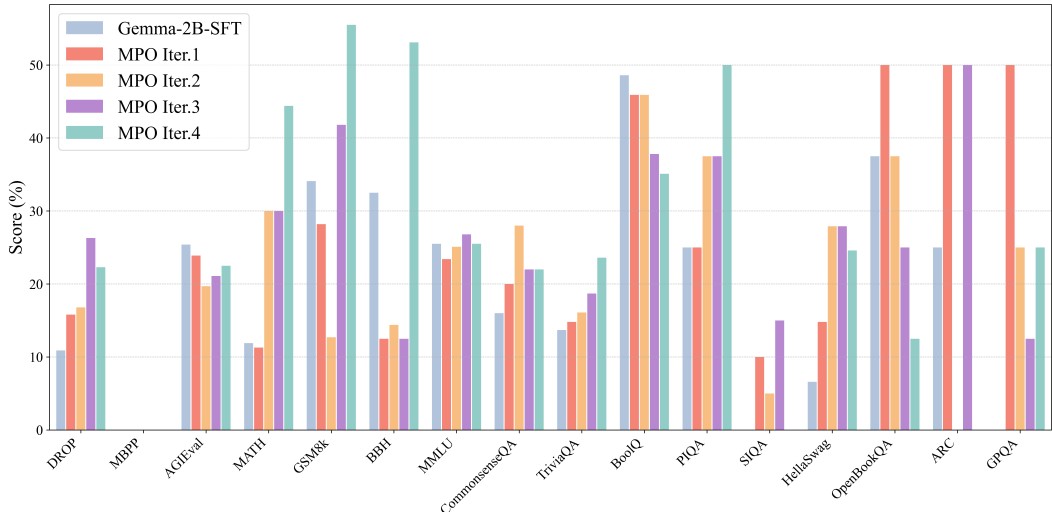

Figure 10: MixEval Hard general ability evalution results for MPO with Gemma-2B-SFT.

on MixEval Hard are shown in Figure 11. From these results, we observe that the performance gains from each round of self-play in MPO-RT are significantly lower than those in MPO, especially in the final two rounds. This result is consistent with the results from the safety alignment evaluation. Additionally, we provide detailed MixEval Hard results for each category, presented in Figure 12. These results indicate that, although theoretically equivalent, directly employ KL regularization on rewards performs much worse than using a KL loss.

| Models/Datasets | PKU-SafeRLHF |
|---|---|
| SFT | 6.37 |
| MPO Iter.1 | -3.76 |
| MPO Iter.2 | -8.96 |
| MPO Iter.3 | **-11.38** |
| MPO-RT Iter.1 | 4.36 |
| MPO-RT Iter.2 | 0.83 |
| MPO-RT Iter.3 | **-4.81** |
| MPO wo. SP | -4.18 |

Table 10: Cost model evaluation results.

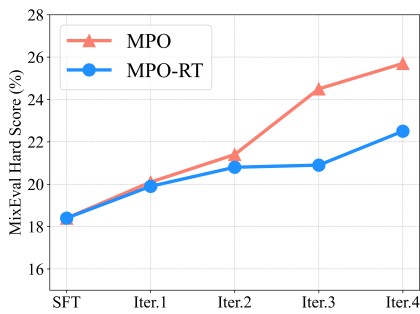

Figure 11: MixEval Hard overall scores for MPO and MPO-RT.

## E PROOFS

### E.1 ADDITIONAL LEMMAS

**Lemma E.1** (Proposition 1, (Munos et al., 2023)). *There exists a unique Nash equilibrium* $(\pi_1^*, \pi_2^*)$ *for the game* $J(\pi_1, \pi_2)$ *defined in* (2) *and* $\pi_1^* = \pi_2^*$.

*Proof.* Since we have that $\mathcal{P}(\pi' > \pi) = 1 - \mathcal{P}(\pi > \pi')$. The minimax game can be repressed as a symmetric two-player game with payoffs of policy $\pi_1$ and $\pi_2$ are defined as

$$R(\pi; \pi') = \mathcal{P}(\pi > \pi') - \alpha D_{\text{KL}}(\pi || \pi_{\text{ref}}),$$

and

$$R(\pi'; \pi) = \mathcal{P}(\pi' > \pi) - \alpha D_{\text{KL}}(\pi' || \pi_{\text{ref}}).$$

We first prove the existence the Nash equilibrium. Since the payoff of the game is concave with respect to $\pi$ and $\pi'$, thus the game processes a Nash equilibrium (Rosen, 1965).

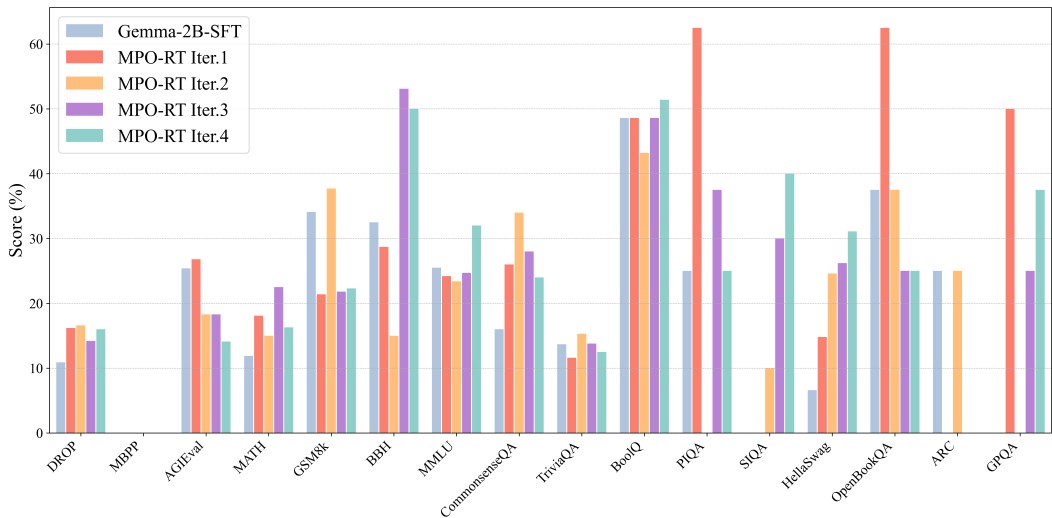

Figure 12: MixEval Hard general ability evalution results for MPO-RT.

For uniqueness, we need to show that the corresponding variational inequality is strictly monotone (Rosen, 1965). Let $\bar{\pi} = [\pi, \pi']$ and $v(\bar{\pi}) = [\nabla_\pi R(\pi; \pi'), \nabla_{\pi'} R(\pi'; \pi)]$. For every Nash equilibrium of the game satisfy

$$v^\top(\bar{\pi}^*)(\bar{\pi}^* - \bar{\pi}) \leqslant 0, \quad \forall \bar{\pi} \in \Pi.$$

The variational inequality is strictly monotone if and only if for every $\bar{\pi}_1$ and $\bar{\pi}_2$, we have

$$(v(\bar{\pi}_1) - v(\bar{\pi}_2))^\top (\bar{\pi}_1 - \bar{\pi}_2) \leqslant 0,$$

with equality only holds at $\bar{\pi}_1 = \bar{\pi}_2$ (Rosen, 1965). We can show this inequality holds by expanding the terms on LHS. For every context $\mathbf{x}$, denote $v(\bar{\pi})(\mathbf{x})$ as the partial derivative for $\mathbf{x}$. We have:

$$v(\bar{\pi})(\mathbf{x}) = \rho(\mathbf{x})[\mathcal{P}(y > \pi' \mid \mathbf{x}) - \alpha \log(\pi/\pi_{\mathrm{ref}} \mid \mathbf{x}) - 1, \mathcal{P}(y > \pi|\mathbf{x}) - \alpha \log(\pi'/\pi_{\mathrm{ref}} \mid \mathbf{x}) - 1],$$

Combining this with Eq E.1 and then exploiting the non-negativity of KL-divergence implies:

$$
\begin{aligned}
(v(\bar{\pi}_1) - v(\bar{\pi}_2))^\top (\bar{\pi}_1 - \bar{\pi}_2) &= \mathcal{P}(\pi_1 > \pi_1') + \mathcal{P}(\pi_1' > \pi_1) + \mathcal{P}(\pi_2 > \pi_2') + \mathcal{P}(\pi_2' > \pi_2) \\
&\quad - \mathcal{P}(\pi_2 > \pi_1') + \mathcal{P}(\pi_1' > \pi_2) + \mathcal{P}(\pi_1 > \pi_2') + \mathcal{P}(\pi_2' > \pi_1) \\
&\quad - \alpha(D_{\mathrm{KL}}(\pi_1|\pi_2) + D_{\mathrm{KL}}(\pi_2|\pi_1) + D_{\mathrm{KL}}(\pi_1'|\pi_2') + D_{\mathrm{KL}}(\pi_2'|\pi_1')) \\
&= -\alpha(D_{\mathrm{KL}}(\pi_1|\pi_2) + D_{\mathrm{KL}}(\pi_2|\pi_1) + D_{\mathrm{KL}}(\pi_1'|\pi_2') + D_{\mathrm{KL}}(\pi_2'|\pi_1')) \leqslant 0.
\end{aligned}
$$

with equality only at $\bar{\pi}_1 = \bar{\pi}_2$. This completes the proof. $\qquad\square$

**Lemma E.2** (Lemma D.3, (Sokota et al., 2022)). *Under the assumptions of Theorem 3.2, we have for all $\pi \in \Pi$,*

$$B_\psi(\pi; \pi^{k+1}) \leqslant B_\psi(\pi; \pi^k) - B_\psi(\pi^{k+1}; \pi^k) + \langle \eta F(\pi^k) + \eta\alpha\nabla g(\pi^{k+1})), \pi - \pi^{k+1}\rangle.$$

*Proof.* Note that

$$\pi^{k+1} \in \underset{\pi\in\Pi}{\arg\min}\, \eta\langle F(\pi^k), \pi\rangle + \eta\alpha g(\pi) + B_\psi(\pi; \pi^k),$$

we have

$$\begin{aligned}
0 &\leqslant \eta\langle F(\pi^k), \pi - \pi^{k+1}\rangle + \eta\alpha\big(g(\pi) - g(\pi^{k+1})\big) + B_\psi(\pi; \pi^k) - B_\psi(\pi^{k+1}; \pi^k) \\
&= \eta\langle F(\pi^k), \pi - \pi^{k+1}\rangle + \eta\alpha\big(g(\pi) - g(\pi^{k+1})\big) + \psi(\pi) - \psi(\pi^{k+1}) - \langle\nabla\psi(\pi^k), \pi - \pi^{k+1}\rangle \\
&\leqslant \eta\langle F(\pi^k), \pi - \pi^{k+1}\rangle + \eta\alpha\langle\nabla g(\pi^{k+1}), \pi - \pi^{k+1}\rangle + \langle\nabla\psi(\pi^{k+1}) - \nabla\psi(\pi^k), \pi - \pi^{k+1}\rangle \\
&= \langle\eta F(\pi^k) + \eta\alpha\nabla g(\pi^{k+1}), \pi - \pi^{k+1}\rangle + \langle\nabla\psi(\pi^{k+1}) - \nabla\psi(\pi^k), \pi - \pi^{k+1}\rangle \\
&= \langle\eta F(\pi^k) + \eta\alpha\nabla g(\pi^{k+1}), \pi - \pi^{k+1}\rangle + B_\psi(\pi; \pi^k) - B_\psi(\pi; \pi^{k+1}) - B_\psi(\pi^{k+1}; \pi^k).
\end{aligned}$$

The first inequality relies on the fact that $\pi^{k+1}$ achieves the minimum value of the objective function. The second inequality results from the first-order optimality condition. Finally, the last equality is derived from either the non-Euclidean prox theorem or the three-point property, as detailed in Bauschke et al. (2003), Beck (2017), Tseng (2008). □

**Lemma E.3** (Lemma D.4, (Sokota et al., 2022)). *Under the assumptions of Theorem 3.2, let $\pi_r^*$ be the solution to $\mathrm{VI}(\Pi, F + \alpha\nabla g)$, then, for $\forall\pi \in \Pi \cap \mathrm{int}\,\mathrm{dom}\psi$, the following equality holds*

$$\langle\eta F(\pi) + \eta\alpha\nabla g(\pi), \pi_r^* - \pi\rangle \leqslant -\eta\alpha\big(B_\psi(\pi; \pi_r^*) + B_\psi(\pi_r^*; \pi)\big).$$

*Proof.* We have
$$\begin{aligned}
\langle\eta F(\pi) + \eta\alpha\nabla g(\pi), \pi_r^* - \pi\rangle &= \langle\eta F(\pi) + \eta\alpha\nabla g(\pi) - \eta F(\pi_r^*) - \eta\alpha\nabla g(\pi_r^*), \pi_r^* - \pi\rangle \\
&\quad + \langle\eta F(\pi_r^*) + \eta\alpha\nabla g(\pi_r^*), \pi_r^* - \pi\rangle \\
&= \langle\eta F(\pi) - \eta F(\pi_r^*), \pi_r^* - \pi\rangle + \eta\alpha\langle\nabla g(\pi) - \nabla g(\pi_r^*), \pi_r^* - \pi\rangle \\
&\quad + \langle\eta F(\pi_r^*) + \eta\alpha\nabla g(\pi_r^*), \pi_r^* - \pi\rangle \\
&\leqslant \eta\alpha\langle\nabla g(\pi) - \nabla g(\pi_r^*), \pi_r^* - \pi\rangle \\
&\leqslant -\eta\alpha\langle\nabla\psi(\pi) - \nabla\psi(\pi_r^*), \pi - \pi_r^*\rangle \\
&= -\eta\alpha\big(B_\psi(\pi; \pi_r^*) + B_\psi(\pi_r^*; \pi)\big).
\end{aligned}$$

The first inequality follows from the monotonicity of the function $F$ and the definition of the solution $\pi_r^*$. The second inequality holds because $g$ is 1-strongly convex relative to $\psi$, and the final equality is derived from the same references as in Lemma E.2. □

**Lemma E.4.** *Under the assumptions of Theorem 3.2, let $\pi_r^*$ be the Nash equilibrium of the regularized game and $\pi^*$ be the Nash equilibrium of the original game. Then the following inequality holds:*

$$\sum_{i\in\mathcal{I}}\langle\nabla_{\pi_i}g_i(\pi_{ri}^*, \pi_{-ri}^*), \pi_{ri}^* - \pi_i^*\rangle \leqslant 0.$$

*Proof.* Since $\pi_r^*$ is the Nash equilibrium of the regularized game, from the first-order optimality condition for $\pi_r^*$, we have for all $\pi \in \Pi$:

$$\sum_{i\in\mathcal{I}}\langle\nabla_{\pi_i}f_i(\pi_{ri}^*, \pi_{-ri}^*) - \alpha\nabla_{\pi_i}g_i(\pi_{ri}^*, \pi_{-ri}^*), \pi_i - \pi_{ri}^*\rangle \leqslant 0.$$

Taking $\pi$ as $\pi^* \in \Pi^*$ and rearranging the inequality, we obtain

$$\begin{aligned}
\sum_{i\in\mathcal{I}}\langle\nabla g_i(\pi_{ri}^*, \pi_{-ri}^*), \pi_{ri}^* - \pi_i^*\rangle &\leqslant \frac{1}{\alpha}\sum_{i\in\mathcal{I}}\langle\nabla_{\pi_i}f_i(\pi_{ri}^*, \pi_{-ri}^*), \pi_{ri}^* - \pi_i^*\rangle \\
&\leqslant \frac{1}{\alpha}\sum_{i\in\mathcal{I}}\langle\nabla_{\pi_i}f_i(\pi_i^*, \pi_{-i}^*), \pi_{ri}^* - \pi_i^*\rangle,
\end{aligned}$$

where the second inequality holds because the game is monotonous. Since $\pi^*$ is the Nash equilibrium of the original game, the first-order optimality condition implies that for all $\pi \in \Pi$,

$$\frac{1}{\alpha}\sum_{i\in\mathcal{I}}\langle\nabla_{\pi_i}f_i(\pi_i^*, \pi_{-i}^*), \pi_i - \pi_i^*\rangle \leqslant 0, \quad \forall\pi \in \Pi.$$

Then,

$$\sum_{i \in \mathcal{I}} \langle \nabla_{\pi_i} g_i(\pi_{ri}^*, \pi_{-ri}^*), \pi_{ri}^* - \pi_i^* \rangle \leqslant \frac{1}{\alpha} \sum_{i \in \mathcal{I}} \langle \nabla_{\pi_i} f_i(\pi_i^*, \pi_{-ri}^*), \pi_{ri}^* - \pi_i^* \rangle \leqslant 0.$$

This concludes the proof. $\qquad\square$

### E.2 PROOF OF THEOREM 3.2

*Proof.*
$$B_\psi(\pi_r^*; \pi^{k+1}) \leqslant B_\psi(\pi_r^*; \pi^k) - B_\psi(\pi^{k+1}; \pi^k) + \langle \eta F(\pi^k) + \eta\alpha \nabla g(\pi^{k+1}), \pi_r^* - \pi^{k+1} \rangle$$
$$= B_\psi(\pi_r^*; \pi^k) - B_\psi(\pi^{k+1}; \pi^k) + \langle \eta F(\pi^k) - \eta F(\pi^{k+1}), \pi_r^* - \pi^{k+1} \rangle + \langle \eta F(\pi^{k+1}) + \eta\alpha \nabla g(\pi^{k+1}), \pi_r^* - \pi^{k+1} \rangle$$

$$\leqslant B_\psi(\pi_r^*; \pi^k) - B_\psi(\pi^{k+1}; \pi^k) + \langle \eta F(\pi^k) - \eta F(\pi^{k+1}), \pi_r^* - \pi^{k+1} \rangle - \eta\alpha \big(B_\psi(\pi^{k+1}; \pi_r^*) + B_\psi(\pi_r^*; \pi^{k+1})\big)$$

$$\leqslant B_\psi(\pi_r^*; \pi^k) - B_\psi(\pi^{k+1}; \pi^k) + \eta L \parallel \pi^k - \pi^{k+1} \parallel \parallel \pi_r^* - \pi^{k+1} \parallel - \eta\alpha \big(B_\psi(\pi^{k+1}; \pi_r^*) + B_\psi(\pi_r^*; \pi^{k+1})\big)$$

$$\leqslant B_\psi(\pi_r^*; \pi^k) - B_\psi(\pi^{k+1}; \pi^k) + \frac{\parallel \pi^k - \pi^{k+1} \parallel^2}{2} + \frac{\eta^2 L^2 \parallel \pi_r^* - \pi^{k+1} \parallel^2}{2} - \eta\alpha \big(B_\psi(\pi^{k+1}; \pi_r^*) + B_\psi(\pi_r^*; \pi^{k+1})\big)$$

$$\leqslant B_\psi(\pi_r^*; \pi^k) + \eta^2 L^2 B_\psi(\pi^{k+1}; \pi_r^*) - \eta\alpha \big(B_\psi(\pi^{k+1}; \pi_r^*) + B_\psi(\pi_r^*; \pi^{k+1})\big)$$
$$\leqslant B_\psi(\pi_r^*; \pi^k) - \eta\alpha B_\psi(\pi_r^*; \pi^{k+1}).$$

The first inequality follows from Lemma E.2, the second inequality from Lemma E.3, the third inequality by the Cauchy-Schwarz inequality and the smoothness of $F$. The fourth inequality is derived from an elementary inequality, and the last inequality follows from the strong convexity of $\psi$ and $B_\psi$. Finally, we obtain

$$B_\psi(\pi_r^*; \pi^{k+1}) \leqslant \frac{1}{1 + \eta\alpha} B_\psi(\pi_r^*; \pi^k).$$

By iteration, we obtain the result in Theorem 3.2. $\qquad\square$

### E.3 PROOF OF LEMMA 3.3

*Proof.* To prove this lemma, we first show that if $\pi_r^{*,n+1} \neq \pi_r^{*,n}$, for $k \geqslant 1$, we have

$$D_{\mathrm{KL}}(\pi^* \| \pi_r^{*,n+1}) < D_{\mathrm{KL}}(\pi^* \| \pi_r^{*,n}).$$

Consider the KL divergence between consecutive iterates. By definition:

$$D_{\mathrm{KL}}(\pi_r^{*,n} \| \pi_r^{*,n+1}) = \sum_{j \in \mathcal{J}} \pi_{rj}^{*,n} \ln \frac{\pi_{rj}^{*,n}}{\pi_{rj}^{*,n+1}}.$$

For any Nash equilibrium $\pi^* \in \Pi^*$, we can write:

$$\sum_{i \in \mathcal{I}} \langle \nabla_i D_{\mathrm{KL}}(\pi_{ri}^{*,n} \| \pi_{ri}^{*,n+1}), \pi_{ri}^{*,n+1} - \pi_i^* \rangle = \sum_{i \in \mathcal{I}} \sum_{j \in \mathcal{J}} (\pi_{ij}^* - \pi_{rij}^{*,n+1}) \frac{\pi_{rij}^{*,n}}{\pi_{rij}^{*,n+1}}$$

$$= I \exp\left( \ln\left( \frac{1}{I} \sum_{i \in \mathcal{I}} \sum_{j \in \mathcal{J}} \pi_{ij}^* \frac{\pi_{rij}^{*,n}}{\pi_{rij}^{*,n+1}} \right) \right) - I$$

$$\geqslant I \exp\left( \frac{1}{I} \sum_{i \in \mathcal{I}} \sum_{j \in \mathcal{J}} \pi_{ij}^* \ln \frac{\pi_{rij}^{*,n}}{\pi_{rij}^{*,n+1}} \right) - I$$

$$= I \exp\left( \frac{1}{I} (D_{\mathrm{KL}}(\pi^* \| \pi_r^{*,n+1}) - D_{\mathrm{KL}}(\pi^* \| \pi_r^{*,n})) \right) - I,$$

where the first equality follows from the gradient of KL divergence, the third inequality follows from Jensen's inequality since $\ln(\cdot)$ is concave, and the last equality is by the definition of KL divergence.

Since $\pi_r^{*,n+1} \neq \pi_r^{*,n}$, we can rearrange to get:

$$D_{\mathrm{KL}}(\pi^* \| \pi_r^{*,n+1}) - D_{\mathrm{KL}}(\pi^* \| \pi_r^{*,n}) < I \ln\left( 1 + \frac{1}{I} \sum_{i \in \mathcal{I}} \langle \nabla_i D_{\mathrm{KL}}(\pi_{ri}^{*,n} \| \pi_{ri}^{*,n+1}), \pi_{ri}^{*,n+1} - \pi_i^* \rangle \right)$$

$$\leqslant \sum_{i \in \mathcal{I}} \langle \nabla_i D_{\mathrm{KL}}(\pi_{ri}^{*,n} \| \pi_{ri}^{*,n+1}), \pi_{ri}^{*,n+1} - \pi_i^* \rangle,$$

where the the second inequality uses the fact that $\ln(x+1) \leqslant x$ for $x > -1$.

Now, we have that

$$D_{\mathrm{KL}}(\pi^* \| \pi_r^{*,n+1}) - D_{\mathrm{KL}}(\pi^* \| \pi_r^{*,n}) \leqslant \sum_{i \in \mathcal{I}} \langle \nabla_i D_{\mathrm{KL}}(\pi_{ri}^{*,n} \| \pi_{ri}^{*,n+1}), \pi_{ri}^{*,n+1} - \pi_i^* \rangle.$$

Since $\pi^*$ is the Nash equilibrium of the original game, $\pi_r^*$ is the Nash equilibrium of the regularized game, and $\pi_r^{*,n+1} \neq \pi_r^{*,n}$, according to Lemma E.4, we have:

$$D_{\mathrm{KL}}(\pi^* \| \pi_r^{*,n+1}) - D_{\mathrm{KL}}(\pi^* \| \pi_r^{*,n}) \leqslant \pi_r^{*,n}) \leqslant \sum_{i \in \mathcal{I}} \langle \nabla_i D_{\mathrm{KL}}(\pi_{ri}^{*,n} \| \pi_{ri}^{*,n+1}), \pi_{ri}^{*,n+1} - \pi_i^* \rangle < 0.$$

Finally, let $\pi_* = \arg\min_{\pi* \in \Pi*} D_{\mathrm{KL}}(\pi^* \| \pi_r^{*,n})$. If $\pi_r^{*,n} \in \Pi \backslash \Pi^*$, then:

$$\min_{\pi* \in \Pi*} D_{\mathrm{KL}}(\pi^* \| \pi_r^{*,n+1}) = D_{\mathrm{KL}}(\pi^* \| \pi_r^{*,n+1}) < D_{\mathrm{KL}}(\pi^* \| \pi_r^{*,n}) = \min_{\pi* \in \Pi*} D_{\mathrm{KL}}(\pi^* \| \pi_r^{*,n}).$$

This shows that the sequence of regularized Nash equilibria strictly approaches the of Nash equilibrium of the original game, completing the proof. $\square$

### E.4 PROOF OF THEOREM 3.4

*Proof.* We will prove that If Lemma 3.3 holds, then the sequence $\{\pi_r^{*,n}\}_{n \geqslant 1}$ converges to $\pi^* \in \Pi^*$ as $n \to \infty$, where $\pi^*$ is the Nash equilibrium of the original game.

Let us begin by noting that since $\min_{\pi* \in \Pi*} D_{\mathrm{KL}}(\pi^* \| \pi_r^{*,n})$ is bounded below by 0, and according to Lemma 3.3, this sequence is strictly decreasing unless $\pi_r^{*,n} \in \Pi^*$, we can conclude that the sequence converges to some value $b \geqslant 0$. We will prove by contradiction that $b = 0$.

Suppose $b > 0$. Let us define:

$$c = \min_{\pi* \in \Pi*} D_{\mathrm{KL}}(\pi^* \| \pi_r^{*,1}),$$

$$\Omega_{b,c} = \{\pi_r \in \Pi \mid b \leqslant \min_{\pi* \in \Pi*} D_{\mathrm{KL}}(\pi^* \| \pi_r) \leqslant c\}.$$

Since $\min_{\pi* \in \Pi*} D_{\mathrm{KL}}(\pi^* \| \pi_r^{*,n})$ decreases monotonically according to Lemma 3.3, we have $\pi_r^{*,n} \in \Omega_{b,c}$ for all $n \geqslant 1$.

First, we prove that $\Omega_{b,c}$ is a compact set. Note that $\min_{\pi* \in \Pi*} D_{\mathrm{KL}}(\pi^* \| \cdot)$ is a continuous function on $\Pi$, and $[b, c]$ is a closed set. Therefore, $\Omega_{b,c}$, being the preimage of $[b, c]$ under a continuous

function, is closed. Furthermore, since $\Pi$ is bounded by assumption, $\Omega_{b,c}$ is also bounded. Thus, $\Omega_{b,c}$ is compact as it is both closed and bounded.

Let $F(\pi_r) : \Pi \to \Pi$ be the function that maps $\pi_r$ to $\pi_r^*$. As proved in Lemma F.7 of Abe et al. (2024), $F$ is a continuous function in our case. When $F$ is continuous, $\min_{\pi^* \in \Pi^*} D_{\mathrm{KL}}(\pi^* \| F(\pi_r)) - \min_{\pi^* \in \Pi^*} D_{\mathrm{KL}}(\pi^* \| \pi_r)$ is also continuous.

Since $\Omega_{b,c}$ is compact and this difference is continuous, there exists a maximum value:

$$\nu = \max_{\pi_r \in \Omega_{b,c}} \{ \min_{\pi^* \in \Pi^*} D_{\mathrm{KL}}(\pi^* \| F(\pi_r)) - \min_{\pi^* \in \Pi^*} D_{\mathrm{KL}}(\pi^* \| \pi_r) \}.$$

From Lemma 3.3, we know that $\nu < 0$. Therefore:

$$\begin{aligned}
\min_{\pi^* \in \Pi^*} D_{\mathrm{KL}}(\pi^* \| \pi_r^{*,N}) = {} & \min_{\pi^* \in \Pi^*} D_{\mathrm{KL}}(\pi^* \| \pi_r^{*,1}) \\
& + \sum_{n=1}^{N} \left( \min_{\pi^* \in \Pi^*} D_{\mathrm{KL}}(\pi^* \| \pi_r^{*,n+1}) - \min_{\pi^* \in \Pi^*} D_{\mathrm{KL}}(\pi^* \| \pi_r^{*,n}) \right) \\
\leqslant {} & c + N\nu.
\end{aligned}$$

This implies that there exists some $N$ large enough such that $c + N\nu \leqslant 0$, contradicting our assumption that $b > 0$. Therefore, we must have $b = 0$ and this proves that the sequence $\{\pi_r^{*,n}\}_{n \geqslant 1}$ converges to $\pi^* \in \Pi^*$, completing our proof. $\qquad\square$

### E.5    PROOF OF THEOREM 3.5

*Proof.* According to the definition of $\pi^{k+1}$, we can directly derive the following equivalence

$$\begin{aligned}
\pi^{k+1} &= \arg\min_{\pi \in \Pi} \eta\big(\langle F(\pi^k), \pi \rangle + \alpha B_\psi(\pi; \pi_{\mathrm{ref}})\big) + B_\psi(\pi; \pi^k) \\
\Leftrightarrow \pi^{k+1} &= \arg\min_{\pi \in \Pi} \langle \eta F(\pi^k) - \eta\alpha\nabla\psi(\pi_{\mathrm{ref}}) - \nabla\psi(\pi^k), \pi \rangle + (1 + \eta\alpha)\psi(\pi) \\
\Leftrightarrow \pi^{k+1} &= \arg\min_{\pi \in \Pi} \langle \frac{\eta F(\pi^k) - \eta\alpha\nabla\psi(\pi_{\mathrm{ref}}) + \nabla\psi(\pi^k)}{1 + \eta\alpha}, \pi \rangle + \psi(\pi) \\
\Leftrightarrow \pi^{k+1} &= \arg\min_{\pi \in \Pi} \langle \frac{\eta F(\pi^k) - \eta\alpha\nabla\psi(\pi_{\mathrm{ref}}) + \eta\alpha\nabla\psi(\pi^k)}{1 + \eta\alpha} - \nabla\psi(\pi^k), \pi \rangle + \psi(\pi) \\
\Leftrightarrow \pi^{k+1} &= \arg\min_{\pi \in \Pi} \langle \bar\eta\big(F(\pi^k) + \alpha\nabla\psi(\pi^k) - \alpha\nabla\psi(\pi_{\mathrm{ref}})\big) - \nabla\psi(\pi^k), \pi \rangle + \psi(\pi) \\
\Leftrightarrow \pi^{k+1} &= \arg\min_{\pi \in \Pi} \langle \bar\eta\big(F(\pi^k) + \alpha\nabla_{\pi^k} B_\psi(\pi^k; \pi_{\mathrm{ref}})\big) - \nabla\psi(\pi^k), \pi \rangle + \psi(\pi) \\
\Leftrightarrow \pi^{k+1} &= \arg\min_{\pi \in \Pi} \bar\eta \langle F(\pi^k) + \alpha\nabla_{\pi^k} B_\psi(\pi^k; \pi_{\mathrm{ref}}), \pi \rangle + B_\psi(\pi; \pi^k).
\end{aligned}$$

This completes the proof. $\qquad\square$

### E.6    PROOF OF LEMMA 3.6

*Proof.* Following the proof of Theorem 3.2, we have
$$D_{\mathrm{KL}}(\pi_r^* \| \pi_r^{k+1}) \leqslant D_{\mathrm{KL}}(\pi_r^* \| \pi_r^k) - \eta\alpha D_{\mathrm{KL}}(\pi_r^* \| \pi_r^{k+1}),$$

for any $k \in \{\tau T_k, \tau T_k + 1, \tau T_k + 2, \ldots, (k+1)T_k - 1\}$,
$$D_{\mathrm{KL}}(\pi_r^* \| \pi_r^{k+1}) \leqslant D_{\mathrm{KL}}(\pi_r^* \| \pi_r^{\tau T_k})(\frac{1}{1 + \eta\alpha})^{k - \tau T_k + 1}.$$

Taking $k = (\tau + 1)T_k - 1$, we have
$$D_{\mathrm{KL}}(\pi_r^* \| \pi_r^{(\tau+1)T_k}) \leqslant D_{\mathrm{KL}}(\pi_r^* \| \pi_r^{\tau T_k})(\frac{1}{1 + \eta\alpha})^{T_k}.$$

Since $\pi_r^\tau = \pi^{\tau T_k}$, we complete our proof. $\qquad\square$

### E.7 PROOF OF THEOREM 3.7

*Proof.* From Lemma 3.6, we have when $T_k \to \infty$, $D_{\mathrm{KL}}(\pi_r^* \| \pi_r^{\tau+1}) \leqslant 0$. Since $D_{\mathrm{KL}}(\pi_r^* \| \pi_r^{\tau+1})$ is never negative, we have $D_{\mathrm{KL}}(\pi_r^* \| \pi_r^{\tau+1}) = 0$. Then, for any $k \in \{\tau T_k, \tau T_k + 1, \tau T_k + 2, \ldots, (k+1)T_k - 1\}$, we obtain our results follows the proof of Theorem 3.4 $\qquad\square$

## F DUALITY GAP AND CONVERGENCE RATE

In this section, we provide an addition theorem for duality gap and show that Theorem 3.2 can be leveraged to guarantee linear convergence of the gap.

**Theorem F.1** (Proposition D.8, (Sokota et al., 2022))**.** *Assume that $g$ is twice continuously differentiable over $\mathrm{int\,dom}\,\psi$, $\Pi$ is bounded, and the assumptions in Theorem 3.2 hold. Let $G = F + \alpha\nabla g$. For all $k \geqslant 1$, the duality gap $\epsilon$ is bounded as*

$$\epsilon(\pi^{k+1}) = \sup_{\pi \in \Pi}\langle G(\pi^k), \pi^k - \pi\rangle \leqslant \mathcal{O}\big((\frac{1}{1+\eta\alpha})^{\frac{k}{2}}\big).$$

*Proof.* From Theorem 3.2, we have that $\{\pi^k\}_{k\geqslant 1} \cup \{\pi^*\}$ is eventually within a closed ball centered at $\pi^*$. So there exists $k'$ and a closed ball $B$ such that $\{\pi^k\}_{k\geqslant k'} \cup \{z^*\} \subseteq \mathrm{int\,dom}\,\psi$. Since $B$ is compact and $\nabla^2 g$ is continuous over $B$, we have that $\nabla^2 g(z)$ is bounded on $B$. Therefore, there exists $L_B$ such that $\|\nabla g(\pi') - \nabla g(\pi)\|^* \leqslant L_B\|\pi - \pi'\|$ for any $\pi, \pi' \in B$. We have that for any $\pi, \pi' \in B$, $\|G(\pi) - G(\pi')\|_* \leqslant \tilde{L}\|\pi - \pi'\|$ for $\tilde{L} = L + \alpha L_B$.

For any $\pi \in \Pi$, we have

$$
\begin{aligned}
\langle G(\pi^*), \pi^{k+1} - \pi\rangle &= \langle G(\pi^*), \pi^{k+1} - \pi\rangle + \langle G(\pi^{k+1}) - G(\pi^*), \pi^{k+1} - \pi\rangle \\
&= \langle G(\pi^*), \pi^* - \pi\rangle + \langle G(\pi^*), \pi^{k+1} - \pi^*\rangle + \langle G(\pi^{k+1}) - G(\pi^*), \pi^{k+1} - \pi\rangle \\
&\leqslant \|G(\pi^*)\|_* \|\pi^{k+1} - \pi^*\| + \tilde{L}\|\pi^{k+1} - \pi^*\|\|\pi^{k+1} - \pi\| \\
&\leqslant \Big(\|G(\pi^*)\|_* + \tilde{L}D\Big)\|\pi^{k+1} - \pi^*\| \\
&\leqslant C\sqrt{B_\psi(\pi^*; \pi^{k+1})} \\
&\leqslant C\left(\sqrt{\frac{1}{1+\eta\alpha}}\right)^k \sqrt{B_\psi(\pi^*; \pi^1)},
\end{aligned}
$$

where $D$ is such that $\max_{\pi, \pi' \in \diamond} \|\pi - \pi'\| \leqslant D$ and $C = \|G(\pi^*)\|_* + \tilde{L}D$.

The first inequality is by the generalized Cauchy-Schwarz inequality and the Lipschitz property of $G$. The second inequality is by boundness of $\Pi$. The third inequality is by the fact that $B_\psi(\pi^*; \pi^k) \geqslant \frac{1}{2}\|\pi^* - \pi^k\|^2$.

$\qquad\square$

**Lemma F.2.** *Let $\{\pi_r^{*,n}\}_{n\geqslant 1}$ be the sequence of NEs of the regularized games, and $\pi_r^\tau$ be the approximation of $\pi_r^{*,n}$ solved via the update rule of (4). Under the assumptions of Theorem 3.2, for any $n \geqslant 1$, if $\pi_r^{*,n} \in \Pi \notin \Pi^*$, we have the following inequality:*

$$B_\psi(\pi^*; \pi_r^{*,n}) \leqslant B_\psi(\pi^*; \pi_r^\tau) - B_\psi(\pi_r^{*,n}; \pi_r^\tau).$$

*Proof.* By the definition of Bregman divergence, we have:

$$B_\psi(\pi^*; \pi_r^{*,n}) - B_\psi(\pi^*; \pi_r^\tau) + B_\psi(\pi_r^{*,n}; \pi_r^\tau) = \sum_{i\in\mathcal{I}}\langle \nabla\psi(\pi_{ri}^{*,n}) - \nabla\psi(\pi_{ri}^\tau), \pi_{ri}^{*,n} - \pi_i^*\rangle.$$

Since $\pi_r^{*,n}$ is the Nash equilibrium of the $n$-th regularized game, by the first-order optimality condition, we have:

$$\sum_{i\in\mathcal{I}}\langle\nabla_{\pi_i}f_i(\pi_{ri}^{*,n})-\alpha\nabla_{\pi_i}B_\psi(\pi_{ri}^{*,n};\pi_{ri}^\tau),\pi_i-\pi_{ri}^{*,n}\rangle\leqslant 0,\quad\forall\pi\in\Pi.$$

Taking $\pi=\pi^*$, we obtain:

$$\sum_{i\in\mathcal{I}}\langle\nabla_{\pi_i}B_\psi(\pi_{ri}^{*,n};\pi_{ri}^\tau),\pi_{ri}^{*,n}-\pi_i^*\rangle\leqslant\frac{1}{\alpha}\sum_{i\in\mathcal{I}}\langle\nabla_{\pi_i}f_i(\pi_{ri}^{*,n}),\pi_{ri}^{*,n}-\pi_i^*\rangle$$

$$\leqslant\frac{1}{\alpha}\sum_{i\in\mathcal{I}}\langle\nabla_{\pi_i}f_i(\pi_i^*),\pi_{ri}^{*,n}-\pi_i^*\rangle,$$

where the second inequality holds because the game is monotonous. Since $\pi^*$ is the Nash equilibrium of the original game, the first-order optimality condition implies that for all $\pi\in\Pi$,

$$\frac{1}{\alpha}\sum_{i\in\mathcal{I}}\langle\nabla_{\pi_i}f_i(\pi_i^*,\pi_{-i}^*),\pi_i-\pi_i^*\rangle\leqslant 0,\quad\forall\pi\in\Pi.$$

Then, taking $\pi=\pi_r^{*,n}$, we have:

$$\sum_{i\in\mathcal{I}}\langle\nabla_{\pi_i}B_\psi(\pi_{ri}^{*,n};\pi_{ri}^\tau),\pi_{ri}^{*,n}-\pi_i^*\rangle=\sum_{i\in\mathcal{I}}\langle\nabla\psi(\pi_{ri}^{*,n})-\nabla\psi(\pi_{ri}^\tau),\pi_{ri}^{*,n}-\pi_i^*\rangle$$

$$\leqslant\frac{1}{\alpha}\sum_{i\in\mathcal{I}}\langle\nabla_{\pi_i}f_i(\pi_i^*),\pi_{ri}^{*,n}-\pi_i^*\rangle\leqslant 0.$$

Thus, we have:

$$B_\psi(\pi^*;\pi_r^{*,n})-B_\psi(\pi^*;\pi_r^\tau)+B_\psi(\pi_r^{*,n};\pi_r^\tau)=\sum_{i\in\mathcal{I}}\langle\nabla\psi(\pi_{ri}^{*,n})-\nabla\psi(\pi_{ri}^\tau),\pi_{ri}^{*,n}-\pi_i^*\rangle\leqslant 0.$$

This completes the proof. $\qquad\square$

**Lemma F.3.** *Under the assumptions of Lemma F.2, the duality gap for $\pi_r^\tau$ is bounded as:*

$$\epsilon(\pi_r^\tau)\leqslant\epsilon(\pi_r^{*,n})+O(\|\pi_r^{*,n}-\pi_r^\tau\|).$$

*Proof.* By the definition of duality gap, we have

$$\epsilon(\pi_r^\tau)=\max_{\pi_i\in\Pi}\sum_{i\in\mathcal{I}}\langle\nabla_{\pi_i}f_i(\pi_{ri}^\tau,\pi_{-ri}^\tau),\pi_i^\tau-\pi_{ri}\rangle$$

$$=\max_{\pi_i\in\Pi}\sum_{i\in\mathcal{I}}\langle\nabla_{\pi_i}f_i(\pi_{ri}^{*,n},\pi_{-ri}^{*,n}),\pi_{ri}^{*,n}-\pi_i\rangle+\max_{\pi_i\in\Pi}\sum_{i\in\mathcal{I}}\langle\nabla_{\pi_i}f_i(\pi_{ri}^\tau,\pi_{-ri}^\tau),\pi_{ri}^\tau-\pi_i\rangle$$

$$-\max_{\pi_i\in\Pi}\sum_{i\in\mathcal{I}}\langle\nabla_{\pi_i}f_i(\pi_{ri}^{*,n},\pi_{-ri}^{*,n}),\pi_{ri}^{*,n}-\pi_i\rangle$$

$$\leqslant\epsilon(\pi_r^{*,n})+\max_{\pi_i\in\Pi}\sum_{i\in\mathcal{I}}\langle\nabla_{\pi_i}f_i(\pi_{ri}^{*,n},\pi_{-ri}^{*,n})-\nabla_{\pi_i}f_i(\pi_{ri}^\tau,\pi_{-ri}^\tau),\pi_i\rangle$$

$$\leqslant\epsilon(\pi_r^{*,n})+L\sum_{i\in\mathcal{I}}\|\pi_{ri}^{*,n}-\pi_{ri}^\tau\|=\epsilon(\pi_r^{*,n})+L\|\pi_r^{*,n}-\pi_r^\tau\|,$$

where the second inequlity follows from the Lipschitz continuity of the gradient with constant $L$. This completes the proof. $\qquad\square$

**Theorem F.4.** *Under the assumptions of Lemma F.2, suppose $\psi$ is $L_\psi$-smooth. Then, for any $N \geqslant 1$, $\forall 1 \leqslant n \leqslant N$, we have*

$$\epsilon(\pi_r^{\tau+1}) \leqslant O(\frac{1}{\sqrt{N}}).$$

*Proof.* From Lemma F.3, we have

$$\epsilon(\pi_r^{\tau+1}) = \epsilon(\pi_r^{*,n}) + L_0 \|\pi_r^{*,n} - \pi_r^{\tau+1}\|,$$

where $L_0$ is a game-dependant constant.

Following the bounding technique for the gap function with tangent residuals (Cai et al., 2022) and the first-order optimality condition for $\pi_r^{*,n}$, we have

$$r^{tan}(\pi_r^{*,n}) \leqslant L_1 \|\pi_r^{*,n} - \pi_r^{\tau}\|,$$

where $r^{tan}(\pi_r^{\tau+1})$ denotes the tangent residual (Cai et al., 2022) of $\pi_r^{*,n}$ and $L_1$ is a constant that depends on the original game. From Lemma 2 of (Cai et al., 2022), for $\forall \pi \in \Pi$, we obtain

$$\epsilon(\pi_r^{*,n}) \leqslant L_2 r^{tan}(\pi_r^{*,n}),$$

where $L_2$ is a game-dependent constant. According to Theorem 3.2, let $\psi = \frac{1}{2}\|\cdot\|^2$, we have

$$\epsilon(\pi_r^{\tau+1}) \leqslant L_1 L_2 \|\pi_r^{*,n} - \pi_r^{\tau}\| + L_0 \|\pi_r^{*,n} - \pi_r^{\tau+1}\|.$$
$$\leqslant L_1 L_2 \|\pi_r^{*,n} - \pi_r^{\tau}\| + L_0 \frac{\|\pi_r^{*,n} - \pi_r^{\tau}\|}{N^c},$$

where $c > 0$ is an arbitrary constant. According to Lemma F.2, we have

$$B_\psi(\pi^*; \pi_r^{*,n}) \leqslant B_\psi(\pi^*; \pi_r^{\tau}) - B_\psi(\pi_r^{*,n}; \pi_r^{\tau})$$
$$\Leftrightarrow 0 \leqslant B_\psi(\pi^*; \pi_r^{\tau}) - B_\psi(\pi_r^{*,n}; \pi_r^{\tau}) - B_\psi(\pi^*; \pi_r^{*,n})$$
$$\Leftrightarrow 0 \leqslant B_\psi(\pi^*; \pi_r^{\tau}) - B_\psi(\pi_r^{*,n}; \pi_r^{\tau}) - B_\psi(\pi^*; \pi_r^{\tau+1}) - B_\psi(\pi_r^{\tau+1}; \pi_r^{*,n})$$
$$+ \langle \nabla\psi(\pi_r^{*,n}) - \nabla\psi(\pi_r^{\tau+1}), \pi^* - \pi_r^{\tau+1} \rangle,$$

where the last line comes from the three-point property of Bregman divergence. For $\langle \nabla\psi(\pi_r^{*,n}) - \nabla\psi(\pi_r^{\tau+1}), \pi^* - \pi_r^{\tau+1} \rangle$, we have

$$\langle \nabla\psi(\pi_r^{*,n}) - \nabla\psi(\pi_r^{\tau+1}), \pi^* - \pi_r^{\tau+1} \rangle \leqslant \|\nabla\psi(\pi_r^{*,n}) - \nabla\psi(\pi_r^{\tau+1})\|\|\pi^* - \pi_r^{\tau+1}\|$$
$$\leqslant \frac{N^c \|\nabla\psi(\pi_r^{*,n}) - \nabla\psi(\pi_r^{\tau+1})\|^2}{2} + \frac{\|\pi^* - \pi_r^{\tau+1}\|^2}{2N^c}$$
$$\leqslant \frac{N^c \|\pi_r^{*,n} - \pi_r^{\tau+1}\|^2}{2} + \frac{\|\pi^* - \pi_r^{\tau+1}\|^2}{2N^c},$$

where the the first inequality follows from the Cauchy-Schwarz inequality, the second inequality follows from $ab \leqslant \rho a^2/2 + b^2/2\rho, \forall \rho > 0$, and the third inequality follows the smoothness of $\psi(\cdot)$. Thus, we have

$$0 \leqslant B_\psi(\pi^*; \pi_r^{\tau}) - B_\psi(\pi_r^{*,n}; \pi_r^{\tau}) - B_\psi(\pi^*; \pi_r^{\tau+1}) - B_\psi(\pi_r^{\tau+1}; \pi_r^{*,n})$$
$$+ \langle \nabla\psi(\pi_r^{*,n}) - \nabla\psi(\pi_r^{\tau+1}), \pi^* - \pi_r^{\tau+1} \rangle$$
$$\leqslant B_\psi(\pi^*; \pi_r^{\tau}) - B_\psi(\pi_r^{*,n}; \pi_r^{\tau}) - B_\psi(\pi^*; \pi_r^{\tau+1}) - B_\psi(\pi_r^{\tau+1}; \pi_r^{*,n})$$
$$+ \frac{N^c \|\pi_r^{*,n} - \pi_r^{\tau+1}\|^2}{2} + \frac{\|\pi^* - \pi_r^{\tau+1}\|^2}{2N^c}$$

Further, we obtain

$$
\begin{aligned}
B_\psi(\pi_r^{*,n}; \pi_r^\tau) \leqslant{}& B_\psi(\pi^*; \pi_r^\tau) - B_\psi(\pi^*; \pi_r^{\tau+1}) - B_\psi(\pi_r^{\tau+1}; \pi_r^{*,n}) \\
& + \frac{\|\pi_r^{*,n} - \pi_r^\tau\|^2}{2} + \frac{\|\pi^* - \pi_r^{\tau+1}\|^2}{2N^c}. \\
\leqslant{}& B_\psi(\pi^*; \pi_r^\tau) - B_\psi(\pi^*; \pi_r^{\tau+1}) + \frac{\|\pi^* - \pi_r^{\tau+1}\|^2}{2N^c}.
\end{aligned}
$$

Then, summing up from $n = 1$ to $N$, we have

$$
\sum_{n=1}^{N} B_\psi(\pi_r^{*,n}; \pi_r^\tau) \leqslant B_\psi(\pi^*; \pi_r^\tau) - B_\psi(\pi^*; \pi_r^{\tau+1}) + \sum_{n=1}^{N} \frac{\|\pi^* - \pi_r^{\tau+1}\|^2}{2N^c}.
$$

Thus, we obtain

$$
\sum_{n=1}^{N} B_\psi(\pi_r^{*,n}; \pi_r^\tau) \leqslant C_3,
$$

where $C_3$ is a game-dependent constant, and this inequality comes from the derivation of (Cevher et al., 2023). Since $c$ is an arbitrary constant, we then have

$$
\begin{aligned}
B_\psi(\pi_r^{*,n}; \pi_r^\tau) ={}& B_\psi(\pi_r^{*,n}; \pi_r^{*,n-1}) + B_\psi(\pi_r^{*,n-1}; \pi_r^\tau) \\
& + \langle \nabla\psi(\pi_r^{*,n}) - \psi(\pi_r^{*,n-1}), \pi_r^{*,n-1} - \pi_r^\tau \rangle \\
\leqslant{}& B_\psi(\pi_r^\tau; \pi_r^{\tau-1}) + B_\psi(\pi_r^{*,n-1}; \pi_r^\tau) + \|\pi_r^{*,n} - \pi_r^{*,n-1}\|\|\pi_r^{*,n-1} - \pi_r^\tau\| \\
\leqslant{}& B_\psi(\pi_r^{*,n-1}; \pi_r^{\tau-1}) + B_\psi(\pi_r^\tau; \pi_r^{*,n-1}) + \|\pi_r^{*,n} - \pi_r^{*,n-1}\|\|\pi_r^{*,n-1} - \pi_r^\tau\| \\
& + B_\psi(\pi_r^{*,n-1}; \pi_r^\tau) + \|\pi_r^\tau - \pi_r^{*,n-1}\|\|\pi_r^{*,n-1} - \pi_r^{\tau-1}\| \\
\leqslant{}& B_\psi(\pi_r^{*,n-1}; \pi_r^{\tau-1}) + 2B_\psi(\pi_r^\tau; \pi_r^{*,n-1}) + \|\pi_r^{*,n} - \pi_r^{*,n-1}\|\|\pi_r^{*,n-1} - \pi_r^\tau\| \\
& + \|\pi_r^\tau - \pi_r^{*,n-1}\|\|\pi_r^{*,n-1} - \pi_r^{\tau-1}\| \\
\leqslant{}& B_\psi(\pi_r^{*,n-1}; \pi_r^{\tau-1}) + \|\pi_r^\tau - \pi_r^{*,n-1}\|^2 + 2C_4\|\pi_r^{*,n-1} - \pi_r^{\tau-1}\| \\
\leqslant{}& B_\psi(\pi_r^{*,n-1}; \pi_r^{\tau-1}) + \frac{1}{N^4}\|\pi_r^\tau - \pi_r^{*,n-1}\|^2 + \frac{2C_4}{N^2}\|\pi_r^{*,n-1} - \pi_r^{\tau-1}\| \\
\leqslant{}& B_\psi(\pi_r^{*,n-1}; \pi_r^{\tau-1}) + \frac{C_4^2}{N^4} + \frac{2C_4^2}{N^2},
\end{aligned}
$$

where $C_4 = \max_{\pi,\pi'\in\Pi} \|\pi - \pi'\|^2$. Then, we have

$$
\begin{aligned}
B_\psi(\pi_r^{*,n}; \pi_r^\tau) &\leqslant B_\psi(\pi_r^{*,n-1}; \pi_r^{\tau-1}) + \frac{C_4^2}{N^4} + \frac{2C_4^2}{N^2}, \\
B_\psi(\pi_r^{*,n}; \pi_r^\tau) &\leqslant B_\psi(\pi_r^{*,n-2}; \pi_r^{\tau-2}) + \frac{2C_4^2}{N^4} + \frac{4C_4^2}{N^2},
\end{aligned}
$$

$$
\cdots
$$

Therefore, we have

$$
\begin{aligned}
& NB_\psi(\pi_r^{*,n}; \pi_r^\tau) \leqslant \sum_{n=1}^{N} B_\psi(\pi_r^{*,n-1}; \pi_r^{\tau-1}) + \frac{C_4^2 N^2}{N^4} + \frac{2C_4^2 K^2}{K^2}, \\
& \Leftrightarrow NB_\psi(\pi_r^{*,n}; \pi_r^\tau) \leqslant C_3 + \frac{C_4^2}{N^2} + 2C_4^2, \\
& \Leftrightarrow \|\pi_r^\tau - \pi_r^{*,n}\| \leqslant O(\frac{1}{\sqrt{N}}),
\end{aligned}
$$

Therefore,

$$
\epsilon(\pi_r^{\tau+1}) \leqslant L_1 L_2 \|\pi_r^{*,n} - \pi_r^\tau\| + L_0 \frac{\|\pi_r^{*,n} - \pi_r^\tau\|}{N^4} \leqslant O(\frac{1}{\sqrt{N}}),
$$

This completes the proof. $\qquad\square$

