# OpenReview forum: "Magnetic Preference Optimization: Achieving Last-iterate Convergence for Language Model Alignment"
_ICLR.cc/2025/Conference — ICLR 2025 Poster_

### Official Review · Reviewer_D3mq · 2024-10-19

**Soundness:** 3
**Presentation:** 2
**Contribution:** 2
**Rating:** 6
**Confidence:** 3

**Summary:**

This paper proposes mirror descent self-play preference optimization (MDSPO), which has a last-iteration convergence to the Nash equilibrium (NE) of the two-player constant-sum normal form game induced by a reinforcement learning from human feedback (RLHF) problem with non-transitive preference. In the regularized magnetic mirror descent setting, the policy converges to the regularized NE at a linear rate, which is faster than the result in previous work. By solving a series of regularized games sequentially, one can have a last-iterate convergence to the NE of the original game. The authors also give practical implementation for the theoretical algorithm and it shows an improvement in several benchmarks.

**Strengths:**

1. This work studies the Nash learning setting, which has broader significance than the Bradley-Terry preference setting as the preference could be arbitrarily non-transitive.
2. The guarantee of last-iteration convergence to NE has practical benefits for large language models (LLMs) as the inference time and stability (low variance) is important.
3. This paper presents the first theoretical result for last-iterate convergence to the NE of the original game, as previous result was for the regularized game.

**Weaknesses:**

1. The convergence rate to the original NE is not studied. The authors should either give an analysis of the convergence rate or explain why such an analysis may be challenging.

**Questions:**

1. The approach described in Section 3.2 seems hard to implement, since we cannot get the exact NE policy $\pi_\text{reg}^{*,n-1}$. What we have from MMD is an approximation of this NE. What will the convergence result be if the approximation error is taken into consideration?
2. In the paper "Nash Learning from Human Feedback" by Munos et. al., the algorithm Nash-MD achieves last-iterate convergence (cf. their Theorem 1) to a regularized NE. Can their algorithm be adapted to the approach in your section Section 3.2 and achieve a last-iterate convergence to the original NE?

---

> ### Author Response · Authors · 2024-11-22
>
> > **W1&Q1: The convergence rate to the original NE is not studied. What will the convergence result be if the approximation error is taken into consideration?**
>
> Thank you for your insightful suggestion to improve our paper.  **We have provided the convergence results when only the approximate NE are learned in Theorem F.4 of the Appendix F.**  Specifically, let $\pi_r^{\tau}$ denote the approximation of NE of the regularized game $\pi_{r}^{*,n}$, with the approximation error measured using the duality gap. **Theorem F.4 demonstrates that duality gap converges at a rate of $O(1/\sqrt{K})$, where $1 \leq n \leq K$.**  We hope this reults adequately addresses your concern.
>
> > **Q2: Can Nash-MD be adapted to the approach in Section 3.2 and achieve a last-iterate convergence to the original NE?**
>
> Thank you for your question. Yes, Nash-MD can indeed be adapted to achieve last-iterate convergence to the original NE following our approach in Section 3.2. **The core insight of our theoretical analysis is that any algorithm capable of achieving last-iterate convergence can be used to solve the regularized game** and the sequence of NE of the regularized games will eventually converge to the NE of the original game, regardless of which algorithm is used as long as it achieves last-iterate convergence.  **Since Nash-MD guarantees last-iterate convergence to the NE of the regularized game, it can be naturally incorporated into our framework.**

---

> > ### Comment · Reviewer_D3mq · 2024-11-22
> >
> > Thanks for the response! I suggest the authors to include the discussion of Q2 into the main text to strengthen the contribution. I'll keep 6.

---

> > > ### Author Response · Authors · 2024-11-23
> > >
> > > Thanks for your constructive suggestion! We have carefully revised the manuscript to include the discussion in the main text.  For your convenience, we have highlighted the changes in the revised manuscript in blue (line 294-299 ).We hope this change helps to better highlight the contribution and improve the clarity of our work. We greatly appreciate your valuable feedback! Please let us know if you have any further suggestions!

---

### Official Review · Reviewer_92Yr · 2024-10-27

**Soundness:** 3
**Presentation:** 2
**Contribution:** 2
**Rating:** 6
**Confidence:** 3

**Summary:**

This paper studies Reinforcement Learning from Human Feedback methods by formulating the problem as finding the Nash Equilibrium of one regularized game. By introducing the Magnetic Mirror Descent approach, they achieve linear last-iterate convergence instead of average-iterate convergence in previous works. Consequently, they provide an algorithm with a theoretical guarantee for convergence, and provide two slightly different ways for implementation (MDSPO and MDSPO-RT) with theoretical insight.

**Strengths:**

1. The extension from SPO to MDSPO algorithm is well-motivated, and the motivation presents well.
2. The experiment is solid, and the performance is good.
3. The idea of periodically updating the magnet policy is interesting, and the theoretical convergence result is provided.
4. In addition to MDSPO, the authors also provide a slightly different algorithm MDSPO-RT, and provide empirical comparisons.

**Weaknesses:**

1. The empirical part:

(1). The connection between theoretical contribution and the empirical implementation is not clear. More explanations about why MMD induces the eq (6) will enhance the paper. For example, the first term on the right side of eq (6) seems unrelated to the policy $pi$ and seems could be ignored, making this equation not similar to the MMD update rule.

(2). In Line 331-333, the paper argues that the choice between MDSPO and MDSPO-RT depends on the requirements of tasks. However, it lacks further guidance on how to make this choice. Additionally, the empirical results only include a task where MDSPO-RT performs worse than MDSPO. Could the authors provide more insight into selecting between the two algorithms?

(3). Since the paper is motivated by the problem of average-iterate convergence, the empirical observation about this phenomenon and how the MDSPO alleviates this problem would significantly strengthen the work. Additionally, empirical comparisons between SPO and MDSPO seem essential for a comprehensive evaluation.

3. The theoretical part:

(1). First, Theorem 3.2 almost follows the same derivation as Theorem 3.4 in [1]. It would be clearer to state explicitly that Theorem 3.2 is a modification of [1] and include the citation prior to presenting it.

(2). The essential theoretical contribution is Lemma 3.3 and Theorem 3.4. However, the proof of these theorems could be polished. For example,  In line 976, what's the term $\langle D_{KL}(\pi_r^{\*,n+1}|| \pi_r^{\*,n}), \pi_r^{\*,n+1}, \pi^\*\rangle $ mean? I assume it's $\langle D_{KL}(\pi_r^{\*,n+1}|| \pi_r^{\*,n}), \pi_r^{\*,n+1}-\pi^\*\rangle.$ Also, in Line 992, it is unclear why the first-order optimality condition can be applied here since the first term $\nabla D_{KL}(\pi_r^{\*,n+1}|| \pi_r^{\*,n})$ doesn't depend on $\pi^\*$. Moreover, in Line 998 and 999, the 3rd and 4th terms in the inequality should involve $\pi_\*$ instead of $\pi^\*$.

(3). In footnote 4, the paper [2] suggests that one could apply optimistic mirror descent (OMD) to achieve a last-iterate guarantee. Why did the authors choose the MMD rather than the OMD? Are there some technical difficulties in directly applying OMD?





[1]. Samuel Sokota, Ryan D’Orazio, J Zico Kolter, Nicolas Loizou, Marc Lanctot, Ioannis Mitliagkas, Noam Brown, and Christian Kroer. A unified approach to reinforcement learning, quantal response equilibria, and two-player zero-sum games.

[2]. Gokul Swamy, Christoph Dann, Rahul Kidambi, Zhiwei Steven Wu, and Alekh Agarwal. A minimaximalist approach to reinforcement learning from human feedback.

**Questions:**

The questions are provided in the weakness part.

---

> ### Author Response · Authors · 2024-11-22
>
> Thank you so much for your appreciation and constructive feedback. Sincerely, we would like to address your concerns as follows.
> > **W1.1: The connection between theoretical contribution and the empirical implementation is not clear.**
>
> Thank you for your question. We apologize for any confusion caused by a typo in our original manuscript. Specifically, in the first term on the right-hand side of eq (6), $y_1$is actually sampled from the current policy. We have corrected this typo in the revised version for clarity. To elaborate, the first term of the MMD objective in eq (4) corresponds to the first-order approximation of the objective function ((for more details, please refer to Chapter 6 of [1])).  Our empirical implementation follows the common approach in prior work (please see eq (4) of [2] and Appendix L of [3]) and is effective in our setting (Figure 1). We hope this clarification helps establish a clearer connection between the theoretical derivations and the practical algorithms used in our work. We sincerely apologize for any misunderstanding caused by the original typo and appreciate your understanding.
>
> > **W1.2: Could the authors provide more insight into selecting between MDSPO and MDSPO-RT?**
>
> Thank you for raising this question. To simplify terminology, we have renamed our algorithm to MPO. The key distinction between MPO and MPO-RT lies in how they handle KL regularization. Specifically, MPO employs a soft constraint on KL divergence through a KL loss, while MPO-RT enforces a hard constraint by directly modifying the reward function, aligning closely with standard RLHF methods.
>
> **Empirically, the choice between MPO and MPO-RT depends on the quality of the preference model and the reference model.** If the preference model is less reliable but the reference model is relatively robust, MPO-RT’s stricter KL constraints can help mitigate the negative effects of the imperfect preference model by keeping the policy closer to the reference. Conversely, if a high-quality preference model is available, MPO typically performs better as its relaxed KL constraints allow the model greater flexibility to explore and optimize within the high-preference regions.
>
> We hope this explanation provides clarity on how to select between the two variants depending on specific application needs and model conditions.
>
> > **W1.3: Empirical observation about the average-iterate convergence phenomenon and comparisons with SPO.**
>
> Thank you for your suggestion. Regarding the empirical observation about the phenomenon and the motivation, please refer to our response to W1 from Reviewer wd8o, where we provide detailed explanations. **As for SPO, it is not well-suited for RLHF scenarios due to the absence of KL regularization with respect to the reference policy.** This lack of regularization causes the policy to deviate rapidly from the reference, ultimately leading to training instability and failure. As a result, SPO is more of a theoretical algorithm rather than a practical solution. Notably, the original SPO paper [4] does not include RLHF experiments, further highlighting its limitations in this context. To demonstrate the practical effectiveness of our proposed algorithm, we have incorporated PPO and Iterative DPO as baselines for comparison. These results are detailed in Appendix D.1. For further discussion on this point, please also refer to our response to W2 from Reviewer wd8o. We hope these additions could address your concern.
>
> > **W2.1&2.2: Citation and proofs of theorems.**
>
> Thank you for pointing out these issues. We have carefully revised our manuscript to ensure proper citation. For the proofs, we have conducted a thorough review and made revisions to improve their clarity and correctness. In particular, we have corrected notational inconsistencies and typos, provided more detailed explanations for critical steps, and enhanced the logical flow of the arguments to ensure they are easier for readers to understand and follow. **Furthermore, we have extended the theoretical contributions by including new results on the convergence rate of our algorithm in Theorem F.2** (please see the reponses to reviewer wd8o), which offers a deeper understanding of its performance and aligns well with the overall theoretical framework. Thank you for your valuable feedback, which has been instrumental in improving the quality of our work.

---

> > ### Author Response · Authors · 2024-11-22
> >
> > > **W2.3: Why did the authors choose the MMD rather than the OMD? Are there some technical difficulties in directly applying OMD?**
> >
> > Thank you for your question. We have indeed explored the possibility of adapting OMD [5] to the RLHF setting, given its well-established theoretical foundation. However,  following the approach outlined in [6], **we found that OMD is not well-suited for RLHF. Specifically, as described in Section 4.2 of [6], implementing OMD requires the agent to maintain a copy of the previous value network to compute optimistic Q-values using the same samples. **Saving a copy of the previous value network at each iteration introduces significant computational overhead.And OMD requires to estimate advantage twice at each iteration [6], making it impractical for RLHF.** Furthermore, our experiments reveal that the deep RL version of OMD performs much poorly compared to MMD, even in small-scale games. In contrast, MMD is not only simple to implement but also achieves superior empirical performance. Additionally, MMD offers valuable insights into the role of KL regularization with respect to the reference policy in RLHF, further solidifying its practical and theoretical advantages in this setting. We hope this explanation clarifies our choice of MMD over OMD in the RLHF setting. Thank you for raising this important point.
> >
> > **References**
> >
> > [1] Orabona, F. (2019). A modern introduction to online learning. arXiv preprint arXiv:1912.13213.
> >
> > [2] Tomar, M., Shani, L., Efroni, Y., & Ghavamzadeh, M. (2020). Mirror descent policy optimization. arXiv preprint arXiv:2005.09814.
> >
> > [3] Samuel Sokota, Ryan D’Orazio, J Zico Kolter, Nicolas Loizou, Marc Lanctot, Ioannis Mitliagkas, Noam Brown, and Christian Kroer. A unified approach to reinforcement learning, quantal response equilibria, and two-player zero-sum games.
> >
> > [4] Swamy, G., Dann, C., Kidambi, R., Wu, Z. S., & Agarwal, A. (2024). A minimaximalist approach to reinforcement learning from human feedback. arXiv preprint arXiv:2401.04056.
> >
> > [5] Mertikopoulos, P., Lecouat, B., Zenati, H., Foo, C. S., Chandrasekhar, V., & Piliouras, G. (2018). Optimistic mirror descent in saddle-point problems: Going the extra (gradient) mile. arXiv preprint arXiv:1807.02629.
> >
> > [6] Moskovitz, T., O’Donoghue, B., Veeriah, V., Flennerhag, S., Singh, S., & Zahavy, T. (2023, July). Reload: Reinforcement learning with optimistic ascent-descent for last-iterate convergence in constrained mdps. In International Conference on Machine Learning (pp. 25303-25336). PMLR.

---

> > > ### Author Response · Authors · 2024-11-25
> > > **Follow-up on Rebuttal and Review Feedback**
> > >
> > > Dear Reviewer 92Yr,
> > >
> > > We sincerely appreciate the time and effort you have devoted to reviewing our work. We understand that your schedule may be quite busy. As the authors-reviewer discussion phase draws to a close, we kindly request your attention to our responses. Our aim is to gain insights into whether our responses effectively address your concerns and to ascertain if there are any additional questions or points you would like to discuss. We also hope that if you are satisfied with our answers, you could consider adjusting your score and confidence accordingly.
> > >
> > > Thank you once again for your thoughtful review and consideration. We greatly value the opportunity to engage with you further.
> > >
> > > Best regards,
> > >
> > > The Authors

---

> > > > ### Comment · Reviewer_92Yr · 2024-11-25
> > > >
> > > > Thanks for the response! The authors solve all of my questions and fix some typos. I am happy to increase my score to 6.
> > > >
> > > > However, there are still a few minor typos remaining in the paper. For example, should the KL divergence terms in Eq.(6) be $D_{KL}(\pi_1||\pi_{r1}^{\*,n-1})$ instead of $D_{KL}(\pi_1||\pi_{r}^{*,n-1})$. The dimension seems inconsistent since $\pi_{r}^{*,n-1}$ is a NE, whereas $\pi_1$ refers to the policy of a single player. Also, for the proof of Lemma 3.3, do we actually have $N=2$?

---

> ### Author Response · Authors · 2024-11-25
>
> Thank you very much for your appreciation and thoughtful feedback! We are delighted to further address your concerns.
>
> For the KL divergence term in Eq.(6), as proved in [1][2], the Nash equilibrium (NE) strategies for both players are unique and identical in our context, i.e.,  $\pi_1^* = \pi_2^* = \pi^*$. This property allows us to use  $\pi^*$  to denote the NE strategies for both players. Thus, for simplicity, we can write the KL divergence terms in Eq. (6) as  $D_{KL}(\pi_1 || \pi_r^{*,n-1})$ . And we have included the proof of this property in Lemma E.1 for completeness.
>
> Regarding your concern about Lemma 3.3, we confirm that  N = 2, as our context involves only two players. To ensure rigor in the proof, we explicitly wrote the expressions for each player in our previous revisions.
>
> In response to the your valuable suggestions, we conducted a thorough review to further improve the clarity of our work and identified a few notational inconsistencies that might cause confusion for readers. Specifically:
>
> 1. In the preliminaries section, we denoted  $\pi^* = (\pi_1^*, \pi_2^*)$  as the NE strategy profile, whereas later,  $\pi^*$  was used to denote the NE strategy for a single player.
>
> 2. We used $N$  to denote the maximum number of players but later used $n$  to refer to the  $n$-th regularized game.
>
> Recognizing that these inconsistencies might mislead the readers, we have made further revisions to the manuscript to refine the notations and proofs. We now use  $\mathcal{I}=${1,2} to denote the set of players, $N$ to denote the number of regularized games and consistently use  $\pi^*$  to represent the NE strategy for a single player to improve clarity. For your convenience, we have highlighted these changes in blue in the preliminaries section.
>
> We sincerely hope these improvements address your concerns and enhance the overall clarity and readability of our work. Thank you again for your appreciation and valuable feedback. Please let us know if you have any further questions or concerns !
>
> **References**
>
> [1] Munos, R., Valko, M., Calandriello, D., Azar, M. G., Rowland, M., Guo, Z. D., ... & Piot, B. (2023). Nash learning from human feedback. arXiv preprint arXiv:2312.00886.
>
> [2] Xiong, W., Dong, H., Ye, C., Wang, Z., Zhong, H., Ji, H., ... & Zhang, T. (2024). Iterative preference learning from human feedback: Bridging theory and practice for rlhf under kl-constraint. In Forty-first International Conference on Machine Learning.

---

> > ### Author Response · Authors · 2024-11-26
> >
> > Dear Reviewer 92Yr,
> >
> > Thank you for your valuable time and thoughtful review of our work. We understand that your schedule may be demanding, and we deeply appreciate your efforts throughout this review process.
> >
> > As we have only one day remaining to revise our manuscript, we kindly ask if you could take a moment to review our latest responses to see whether they adequately address your concerns. If you have any additional questions, we would be more than happy to engage further with you and revise our manuscript to ensure your concerns are fully resolved.
> >
> > Once again, thank you for your insightful suggestions and consideration. We look forward to your valuable feedback.
> >
> > Best regards,
> >
> > The Authors

---

> > > ### Comment · Reviewer_92Yr · 2024-11-26
> > >
> > > Thanks for the detailed response. I have increased my score to 6.

---

> > > > ### Author Response · Authors · 2024-11-26
> > > >
> > > > Thanks for taking the time to review our response and thoughtful review! We deeply appreciate your efforts!

---

### Official Review · Reviewer_wd8o · 2024-11-02

**Soundness:** 3
**Presentation:** 2
**Contribution:** 3
**Rating:** 8
**Confidence:** 3

**Summary:**

The paper presents the Magnetic Mirror Descent Self-Play Preference Optimization (MDSPO) framework, a novel approach in the field of Reinforcement Learning from Human Feedback (RLHF). It addresses the limitations of the Bradley-Terry model by reframing preference learning as a two-player constant-sum game and aims to converge to the Nash equilibrium (NE) of this game. MDSPO introduces a magnetic term to achieve linear convergence to the NE of the regularized game and provides theoretical guarantees for last-iterate convergence to the NE of the original game. The empirical results demonstrate significant improvements compared to the SFT method under the MixEval-Hard benchmark.

**Strengths:**

1. By using the proposed magnetic mirror descent (MMD) method, this work achieves the last iteration convergence guarantee for the NE of the game under the general preference setting, which is better than previous works.
2. The proposed MDSPO method achieves a great performance compared to the SFT method.

**Weaknesses:**

1. The motivation for introducing the magnetic term in MMD is currently unclear. It would be helpful to discuss the technique challenge to achieve last-iteration convergence when using mirror descent. After that, this work can show the reason why the MMD method can solve the above problem and achieve great performance, which would deepen the understanding of MMD.
2. For the experiment part, this work uses the SFT method as the baseline. It would be better to add additional experiments such as the RLHF with PPO to do a more comprehensive comparsion.

**Questions:**

Please see the Weakness part.

Typo: It should be $y_{<t}$ instead of $ y_{<k}$ in line 116.

---

> ### Author Response · Authors · 2024-11-22
>
> Thank you so much for your appreciation and constructive feedback. Sincerely, we would like to address your concerns as follows.
>
> >**W1:The motivation for introducing the magnetic term in MMD is currently unclear. It would be helpful to discuss the technique challenge to achieve last-iteration convergence when using mirror descent.**
>
> Thanks for your valuable feedback. We acknowledge that the motivation for introducing the magnetic term[1] in Magnetic Mirror Descent (MMD) could be clarified further. **To be straightforward and concise, the magnetic term in MMD facilitates a more stable convergence of a single strategy towards NE. This signifies that we can achieve preference alignment with just a single model, which is of crucial significance for improving the training stability for large models and mitigating the inference costs.** Then we will elaborate on this in detail. We have incorporated additional explanations and figures (Figure 1 and Figure 3) in our manuscript to help readers better understand our approach. For your convenience, we have highlighted these changes in the revised manuscript in blue.
>
> As demonstrated in Figure 3, which shows the trajectories of MD and MMD in solving a simple game (e.g., $f(x, y) = (x-1)(y-1)$), the traditional Mirror Descent (MD) algorithm achieves average-iterate convergence to the NE. However, the last-iterate policy of MD (depicted by the outer blue curve in the figure) oscillates around the NE (marked as the red star at the center). **This oscillatory behavior makes it challenging to directly apply MD to practical RLHF scenarios, as the final policy often deviates significantly from the NE.** To approximate the NE, additional mechanisms like averaging are required, which increase storage and computational overhead.
>
> To overcome this challenge, the magnetic term in MMD introduces a regularization force that attracts policy updates toward a reference policy while maintaining progress toward the NE. **This magnetic attraction stabilizes the updates, reducing oscillations and enabling linear last-iterate convergence to the NE of the regularized game.** When the reference policy is a uniform policy, this term reduces to entropy regularization. Our experiments on Kuhn Poker (shown in Figure 1) further demonstrate the limitations of MD-based deep RL methods, such as PPO and SAC (as suggested in SPO [2]). The duality gap in the figure measures the distance to the NE, with lower values indicating closer to the NE. MD-based methods exhibit poor performance in achieving last-iterate convergence to the NE, underscoring the necessity of regularization.
>
> However, as shown in Figure 3, **MMD only achieves last-iterate convergence only to the NE of the regularized game, rather than the original game**, which is the ultimate alignment objective (this corresponds to the Minmax Winner solution concept in social choice theory [3]). Furthermore, **larger regularization strengths (larger $\alpha$) cause the converged policy to deviate further from the original alignment objective.** To address this limitation, **our results demonstrate that we can update the reference policy periodically to guide the policy to converge to the original NE.** We provide detailed theoretical analysis of this approach, which ensures that our method achieves last-iterate convergence to the original NE while balancing stability and computational efficiency.
>
> > **W2: For the experiment part, this work uses the SFT method as the baseline. It would be better to add additional experiments such as the RLHF with PPO to do a more comprehensive comparsion.**
>
> Thank you for your valuable suggestion. To address your concern,  **we have extended our experimental evaluation by including both PPO and Iterative DPO[2] as additional baselines for comparison. The updated results and discussion can be found in Appendix D.1**, where we detail the experimental setups and methodologies to ensure a fair comparison. The results in Figure 7 demonstrate that MPO (we renamed our proposed algorithm as MPO for simplicity) outperforms both PPO and Iterative DPO.
> We appreciate your suggestion and we are looking forward to your further feedback.
>
> We sincerely appreciate your suggestion and hope that these responses adequately address your concerns.
>
> **References**
>
> [1] Samuel Sokota, Ryan D’Orazio, J Zico Kolter, Nicolas Loizou, Marc Lanctot, Ioannis Mitliagkas, Noam Brown, and Christian Kroer. A unified approach to reinforcement learning, quantal response equilibria, and two-player zero-sum games.
>
> [2] Swamy, G., Dann, C., Kidambi, R., Wu, Z. S., & Agarwal, A. (2024). A minimaximalist approach to reinforcement learning from human feedback. arXiv preprint arXiv:2401.04056.
>
> [3] Dong, H., Xiong, W., Pang, B., Wang, H., Zhao, H., Zhou, Y., ... & Zhang, T. (2024). Rlhf workflow: From reward modeling to online rlhf. arXiv preprint arXiv:2405.07863.

---

> > ### Comment · Reviewer_wd8o · 2024-11-26
> >
> > Thanks for the detailed response. As all my concerns are addressed, I will keep my score.

---

> > > ### Author Response · Authors · 2024-11-26
> > >
> > > Thank you for acknowledging our response! We appreciate your effort and valuable feedback!

---

### Official Review · Reviewer_8BmD · 2024-11-04

**Soundness:** 3
**Presentation:** 2
**Contribution:** 3
**Rating:** 6
**Confidence:** 3

**Summary:**

In this paper the authors study the problem of Reinforcement Learning from Human Feedback. As it was shown in bibliography this problem can be seen as a two-player constant sum game with Nash equilibrium as a solution. In this paper the authors study the last-iterate convergence to a Nash equilibrium in this context giving new dynamics for this. They show last-iterate convergence and give experiments that show evidence that there is improvement of the problem with their algorithm.

**Strengths:**

I think that it is nice to see how different areas of research are combined and results of the one area can be used to the other. In this direction the authors study/prove the last-iterate convergence to a Nash equilibrium of a game in their context, giving new techniques and ideas. Furthermore, this paper shows another example of the importance of last-iterate convergence to Nash equilibria in constant sum games.

**Weaknesses:**

I think the presentation of paper has space of improvement, especially in the technical/mathematical part.

**Questions:**

In 145 line it is stated that the constant sum game has unique and symmetric Nash equilibrium in this context (which is not true for any constant-sum game in general). Do similar results hold without having the uniqueness assumption? In what exact point in the proofs the uniqueness of the Nash equilibrium is needed?

---

> ### Author Response · Authors · 2024-11-22
>
> Thank you so much for your appreciation and constructive feedback. Sincerely, we would like to address your concerns as follows.
>
> > **W1:Presentation of paper has space of improvement, especially in the technical/mathematical part.**
>
> Thank you for your valuable suggestion. We have thoroughly revised the paper to enhance its readability and overall presentation. For your convenience, we have highlighted significant changes in the revised manuscript in blue.
>
> To improve clarity, **we refined the expressions throughout the paper and simplified the notation to make it more accessible.** Additionally, we incorporated extra figures (Figure 1 and Figure 3 in our revised manuscript) to provide an intuitive understanding of our motivation and key concepts. To complement the main text, **we expanded the appendix with additional results and details of our algorithms and experimental setups**, ensuring that readers can gain a comprehensive understanding of our methods.
>
> For the technical part, **we introduced additional explanations around the key theorems to provide high-level insights**, making the results easier for readers to understand. Furthermore, we carefully revised the proofs, adding necessary clarifications at critical steps to improve their transparency and logical flow.
>
> These improvements aim to make the paper more accessible, easier to read, and more engaging. We sincerely hope these revisions address your concerns and enhance your overall experience with the work.
>
> > **Q1: Do similar results hold without having the uniqueness assumption? In what exact point in the proofs the uniqueness of the Nash equilibrium is needed?**
>
> We appreciate your careful reading and thoughtful questions. **We want to clarify that our theoretical results do not rely on the uniqueness assumption due to the key condition required in our proof is the mild monotonicity (where the “<=” holds in the following formula) on the gradient of payoff functions, rather than strict monotonicity (where the “<” strictly hold):
>   $\sum_{i=1}^N \langle \nabla_{\pi_{i}} f(\pi_i, \pi_{-i})-\nabla_{\pi_{i}} f(\pi_i^{\prime}, \pi_{-i}^{\prime}), \pi_i - \pi_i^{\prime}\rangle \leq 0, \forall \pi, \pi^{\prime} \in \Pi$.
> This mild monotonicity condition does not require the uniqueness of Nash equilibrium, which is sufficient for our convergence analysis,** while the uniqueness of Nash equilibrium implies strict monotonicity (strict inequality).
>
> This conditional is mainly used to derive Lemma E.3 (line 1346), which is necessary for deriving Lemma 3.3 and Theorem 3.4. And we don't need strict monotonicity here. Therefore, as long as the mild monotonicity condition is satisfied, our theoretical framework naturally extends to cases with multiple Nash equilibriums. It is just that, in our specific scenario, the Nash equilibrium is unique as proved in[1][2], and we have included the proof in Lemma E.1 for completeness.
>
> Thank you again for helping us improve our theoretical clarity. Please let us know if you have any other suggestions.
>
> **References**
>
> [1] Munos, R., Valko, M., Calandriello, D., Azar, M. G., Rowland, M., Guo, Z. D., ... & Piot, B. (2023). Nash learning from human feedback. arXiv preprint arXiv:2312.00886.
>
> [2] Xiong, W., Dong, H., Ye, C., Wang, Z., Zhong, H., Ji, H., ... & Zhang, T. (2024). Iterative preference learning from human feedback: Bridging theory and practice for rlhf under kl-constraint. In Forty-first International Conference on Machine Learning.

---

> > ### Author Response · Authors · 2024-11-25
> > **Follow-up on Rebuttal and Review Feedback**
> >
> > Dear Reviewer 8BmD,
> >
> > We sincerely appreciate the time and effort you have devoted to reviewing our work. We understand that your schedule may be quite busy. As the authors-reviewer discussion phase draws to a close, we kindly request your attention to our responses. Our aim is to gain insights into whether our responses effectively address your concerns and to ascertain if there are any additional questions or points you would like to discuss. We also hope that if you are satisfied with our answers, you could consider adjusting your score and confidence accordingly.
> >
> > Thank you once again for your thoughtful review and consideration. We greatly value the opportunity to engage with you further.
> >
> > Best regards,
> >
> > The Authors

---

> > > ### Comment · Reviewer_8BmD · 2024-11-26
> > > **Response to Authors**
> > >
> > > Thank you for your detailed answers. I will maintain my initial score on the review.

---

> > > > ### Author Response · Authors · 2024-11-26
> > > >
> > > > Thank you for your thoughtful review and consideration!

---

### Author Response · Authors · 2024-11-22

We would like to sincerely thank all reviewers for their valuable and constructive comments, which have been instrumental in guiding our revisions and improving the quality of our paper. In the most respectful manner, we summarize our key updates to the paper as follows:
1.  **We have conducted a thorough revision of the manuscript to enhance its readability and presentation.** Additional explanations and figures have been included to make the insights of our work more accessible and easier to understand. Furthermore, **we have provided more implementation details of our proposed algorithm in the appendix.** (Reviewer 8BmD, Reviewer 92Yr)
2. **We have included the convergence results for scenarios where only approximate Nash Equilibria are learned**, presented in Theorem F.4 of Appendix F. Additionally, we conducted a comprehensive review and made revisions to our proofs to ensure clarity and correctness.(Reviewer wd8o, Reviewer 92Yr)
3. To provide a more comprehensive comparison, **we have incorporated additional baseline methods in Appendix D.1, including PPO and Iterative DPO.** These additions further demonstrate the effectiveness of our proposed algorithm. (Reviewer wd8o, Reviewer 92Yr)

For the reviewers’ convenience, we have highlighted some of the significant changes in the revised manuscript in blue. Other questions and concerns raised by reviewers have been addressed in detail in our individual responses. We hope our revisions and explanations address your comments satisfactorily and look forward to any further feedback you may have.

---

### Meta-Review · Area_Chair_Udn9 · 2024-12-21

**Metareview:**

In this paper the authors study the problem of Reinforcement Learning from Human Feedback which has many applications in boosting the performance of LLMs. One can cast this problem as a two-player constant sum game with Nash equilibrium as a solution. In this paper the authors study the last-iterate convergence to a Nash equilibrium in this context giving new dynamics for this. All the reviewers are positive or slightly positive about the paper and the AC recommends acceptance.

**Additional Comments On Reviewer Discussion:**

A couple of reviewers increased their score after the rebuttal. The authors did a good job addressing the comments.

---

### Decision · Program_Chairs · 2025-01-22

Accept (Poster)